# The Ube2m-Rbx1 neddylation-Cullin-RING-Ligase proteins are essential for the maintenance of Regulatory T cell fitness

Di Wu [1,2,3], Haomin Li [4], Mingwei Liu [5], Jun Qin[5] & Yi Sun [1,2,3] ✉

Neddylation-mediated activation of Cullin-RING E3 Ligases (CRLs) are necessary for the degradation of specific immune regulatory proteins. However, little is known about how these processes govern the function of regulatory T (Treg) cells. Here we show that mice with Treg cell-specific deletion of Rbx1, a dual E3 for both neddylation and ubiquitylation by CRLs, develop an early-onset fatal inflammatory disorder, characterized by disrupted Treg cell homeostasis and suppressive functions. Specifically, Rbx1 is essential for the maintenance of an effector Treg cell subpopulation, and regulates several inflammatory pathways. Similar but less severe phenotypes are observed in mice having Ube2m, a neddylation E2 conjugation enzyme, deleted in their Treg cells. Interestingly, Treg-specific deletion of Rbx2/Sag or Ube2f, components of a similar but distinct neddylation-CRL complex, yields no obvious phenotype. Thus, our work demonstrates that the Ube2m-Rbx1 axis is specifically required for intrinsic regulatory processes in Treg cells; and that Rbx1 might also play Ube2m-independent roles in maintaining the fitness of Treg cells, suggesting a layer of complexity in neddylation-dependent activation of CRLs.

[1] Cancer Institute of the Second Affiliated Hospital and Institute of Translational Medicine, Zhejiang University School of Medicine, Hangzhou, Zhejiang 310029, China. [2] Cancer Center, Zhejiang University, Hangzhou, Zhejiang 310058, China. [3] Research Center for Life Science and Human Health, Binjiang Institute of Zhejiang University, Hangzhou, Zhejiang 310053, China. [4] Children's Hospital, Zhejiang University School of Medicine, Hangzhou, Zhejiang 310003, China. [5] State Key Laboratory of Proteomics, Beijing Proteome Research Center, National Center for Protein Sciences (Beijing) and Institute of Lifeomics, Beijing 102206, China. ✉email: yisun@zju.edu.cn

CRLs are a family of multi-unit E3 ubiquitin ligases complex, consisting of 4 subunits: a scaffold cullin with 8 family members, an adaptor with many members, a substrate receptor with hundreds of members, and a RING component with two family members, RING-box 1 (Rbx1/Roc1) and RING-box 2 (Rbx2/Roc2/Sag)[1]. The two members of RING family act as the catalytic subunit: Rbx1 couples with CRLs1–4, whereas RING-box 2 (Rbx2/Sag) couples mainly with CRL5[2]. By promoting ubiquitylation of roughly 20% of cellular proteins doomed for proteasome degradation[3], CRLs, as the largest family of E3 ubiquitin ligases, play an essential role in the regulation of apoptosis, cell cycle progression, DNA replication, DNA damage response and repair, gene transcription, genome integrity, signal transduction, stress responsiveness, and tumorigenesis[4,5].

Activity of CRLs requires neddylation on their cullin subunit, catalyzed by neddylation enzyme cascades, including E1 NEDD8 activation enzyme (NAE), a heterodimer of NAE1/APPBP1 and UBA3/NAEβ; E2 NEDD8 conjugating enzyme with two family members UBE2M and UBE2F, and several E3 NEDD8 ligases. Both Rbx1 and Rbx2 can also serve as co-E3 NEDD8 ligase[6]. It is well-established that UBE2M couples with RBX1 E3 to promote neddylation of cullins 1–4, whereas UBE2F couples with RBX2/SAG E3 for cullin-5 neddylation[2].

The total knockout studies from our laboratory have previously shown that Rbx1[7] and Rbx2/Sag[8], are functionally non-redundant and both required for mouse embryogenesis. Our recent study showed that RBX1 binds to CDC34/UBCH2C to promote ubiquitylation of large number of substrates via the K48 linkage, whereas RBX2 binds to UBE2C/UBE2S to promote ubiquitylation of different set of substrates via the K11 linkage for proteasome degradation[9,10]. Recent studies showed that RBX1 also binds to UBCH5B/UBE2D2[11], UBCH5C/UBE2D3[12] E2s and ARIH1 E3[13], whereas RBX2 binds to ARIH2 E3[13], further demonstrating a new layer of difference between two family members.

A number of studies, using various cellular and mouse models, have shown that neddylaiton and CRLs regulate immune system via promoting ubiquitylation and degradation of specific immunological substrates. Examples include CRL1 regulation of NF-κB pathway[14] and inflammatory cytokines[15], CRL2$^{Vhl}$ regulation of Treg cells via targeting Hif1α[16], CRL3$^{SPOP}$ regulation of PD-L1[17], CRL5$^{Socs}$ controlling of JAK/STAT pathway[18], and Ube2m/Ubc12 regulation of efficient Th1 and Th2 differentiation of CD4$^+$ T cells[19].

While no conditional Rbx1 knockout mouse study was reported so far, our previous studies showed that T-cell specific Rbx2/Sag knockout (via Lck-Cre) significantly decreased T cell activation, proliferation, and T-effector cytokine release, although mice are viable[20]; whereas Sag knockout in myeloid lineage (via LysM-Cre) increased serum levels of proinflammatory cytokines and enhanced mortality in response to LPS[21]. We have also shown the key role of neddylation in functional regulation dendritic cells (DC), given that neddylation inhibitor, MLN4924 suppressed the release of proinflammatory cytokines by DCs in response to various stimuli, and suppressed the ability of DCs to stimulate both murine and human allogeneic T cell responses[22].

Regulatory T cells are specialized immunosuppressive CD4$^+$-T lymphocytes, that play pivotal roles in maintaining immune homeostasis in vivo, illustrated by the early onset fatal autoimmune disorders upon depletion of Treg cells[23]. Transcription factor Forkhead box P3 (Foxp3) is the master marker to distinguish the Treg cells from other CD4$^+$-T cells[24–26], discovery of additional key regulators of Treg cells is required to gain a better understanding of fundamental biology of Treg cells. So far, there is no systematic study using mouse conditional knockout models to elucidate the physiological role of CRLs (via Rbx1/Rbx2) and neddylation (via Ube2m/Ube2f) in regulation of functions and survival of Treg cells for proper maintenance of homeostasis in immune system, even though both neddylation/CRLs and Treg cells play a fundamental role in immune modulation.

In this study, we investigate whether and how Rbx1-Rbx2/CRLs and Ube2m-Ube2f/neddylation regulate the homeostasis and function of Treg cells, and find that while mice with a Treg-specific deletion of Rbx2/Sag or Ube2f show no obvious phenotypes, mice with Rbx1 deletion in Treg cells develop an early-onset fetal inflammatory disorder and death at day ~25 after birth (~p25), with impaired suppressive functions and disrupted homeostasis of Treg cells, indicating Rbx1 as a prominent regulator of Treg cells. Moreover, mice with Ube2m deletion in Treg cells also suffers from inflammatory disorders, but to a much lesser severity with a 50% of death rate at ~150 days of age. The immune suppressive function of Treg cells deficient of either Rbx1 or Ube2m are severely or moderately compromised, respectively. Unbiased transcriptome comparison between Rbx1-deficient and Ube2m-deficient Treg cells reveals an overlapping but also unique changes in the signaling pathways controlling the inflammatory responses with the former showing greater alterations. Our study suggests that targeting CRL-neddylation could be an effective approach for the treatment of human diseases with over-activated Treg cells.

## Results

**Early-onset fatal inflammation in Foxp3$^{cre}$;Rbx1$^{fl/fl}$ mice.** To investigate the role of CRLs in the regulation of Treg cells in vivo, we generated conditional knockout mouse models with inactivation of Rbx1 and Rbx2/Sag individually in Treg cells, driven by Foxp3$^{YFP-cre}$ (Foxp3$^{cre}$)[27,28]. The compound mice were designated as Foxp3$^{cre}$;Rbx1$^{fl/fl}$ or Foxp3$^{cre}$;Sag$^{fl/fl}$ mice, respectively. Although Sag was expressed in wt Treg cells, Treg-specific depletion of Sag (Supplementary Fig. 1a) does not obviously impair Treg cell function and survival at the steady state (Supplementary Fig. 1b–d). Specifically, compared to Foxp3$^{cre}$ controls (wild-type/wt), Foxp3$^{cre}$;Sag$^{fl/fl}$ mice were viable and healthy without obvious alterations in activation markers of CD4$^+$Foxp3$^-$ T cells (conventional T cells, or Tcon cells) (Supplementary Fig. 1b), the Tcon cell proliferation rate (Supplementary Fig. 1c), and the Treg/CD4$^+$ ratio (Supplementary Fig. 1d), suggesting that the role of Sag in Treg cells at the steady state is, if any, not obvious.

Strikingly, deletion of Rbx1 in Treg cells (Supplementary Fig. 2a) was early-onset fatal. Compared to wt controls, Foxp3$^{cre}$;Rbx1$^{fl/fl}$ mice showed an altered appearance as early as day 13–15 after birth (p13–15), with collapsed ears, festered skin, and reduced body size (Fig. 1a). The mice continued to lose weight dramatically after p15 (Fig. 1b) with a 50% or 100% of death rate at p25 or p37, respectively (Fig. 1c). Autopsy examination revealed that Foxp3$^{cre}$;Rbx1$^{fl/fl}$ mice had swollen lymph nodes and spleens (Fig. 1d, e), with lymphocyte infiltration into multiple organs, including the skin, lung, stomach, liver, kidney, and colon (Supplementary Fig. 2b). The Foxp3$^{cre}$;Rbx1$^{fl/fl}$ mice also had a decreased CD4$^+$/CD8$^+$ T-cell ratio (Supplementary Fig. 3a, b) and a significantly increased proportion of effector/memory T cells (CD44$^{hi}$CD62L$^{lo}$) among populations of Tcon cells (Fig. 1f and Supplementary Fig. 3c). Beyond CD4$^+$-T cells, multiple types of immune cells, including CD8$^+$ cytotoxic T cells, macrophages, dendritic cells, B lymphocytes, were all dramatically activated in Foxp3$^{cre}$;Rbx1$^{fl/fl}$ mice (Supplementary Fig. 4), indicating a typical phenotype of autoimmune disease. Such robust inflammation observed in Foxp3$^{cre}$;Rbx1$^{fl/fl}$ mice is reminiscent of Treg-deficient mice[23] or mice with loss-of-function mutations in the Foxp3 gene[29], suggesting Rbx1 is absolutely essential for Treg cells in vivo.

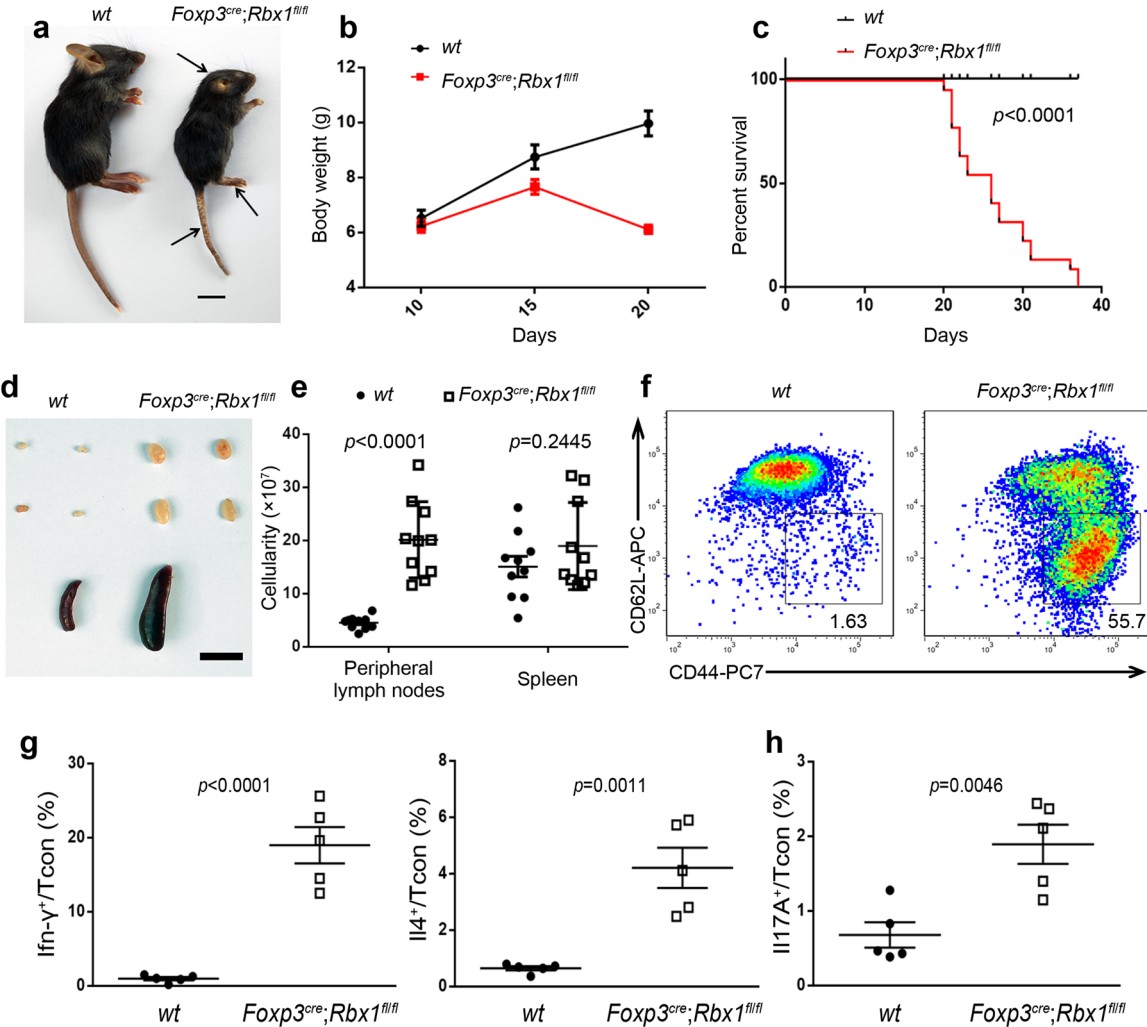

**Fig. 1 *Rbx1* deletion in Treg cells leads to an early-onset fatal inflammatory disorder. a** Representative images of *wt* and *Foxp3*$^{cre}$;*Rbx1*$^{fl/fl}$ mice (p20, male, scale bar = 1 cm). **b** Gross body weight of *wt* and *Foxp3*$^{cre}$;*Rbx1*$^{fl/fl}$ mice (p19–23, n = 10 animals, for gender information see "source data"). **c** Survival curve of *wt* and *Foxp3*$^{cre}$;*Rbx1*$^{fl/fl}$ mice (n = 21 animals from both male and female mice, p < 0.0001). **d** Representative images of the peripheral lymph nodes (top) and spleen (bottom) from *wt* and *Foxp3*$^{cre}$;*Rbx1*$^{fl/fl}$ mice (p22, male, scale bar = 1 cm). **e** Total cell numbers in peripheral lymph nodes and spleen from *wt* and *Foxp3*$^{cre}$;*Rbx1*$^{fl/fl}$ mice (p19–23, n = 10 biologically independent samples from both male and female mice, p < 0.0001 for left, and p = 0.2445 for right). **f** Expression of CD44 and CD62L in Tcon cells from peripheral lymph nodes of *wt* and *Foxp3*$^{cre}$;*Rbx1*$^{fl/fl}$ mice (p21, male). **g** The levels of Ifn-γ and Il-4 in Tcon cells from peripheral lymph nodes of *wt* and *Foxp3*$^{cre}$;*Rbx1*$^{fl/fl}$ mice (p19–23, n = 5 biologically independent samples from both male and female mice, p < 0.0001 for left, and p = 0.0011 for right). **h** The levels of Il-17A in Tcon cells from peripheral lymph nodes of *wt* and *Foxp3*$^{cre}$;*Rbx1*$^{fl/fl}$ mice (p19–23, n = 5 biologically independent samples from both male and female mice, p = 0.0046). All error bars represent the SEM, data are presented as mean values +/− SEM. The p values were calculated by Mann–Whitney test. Source data are provided as a Source Data file.

The levels of Ifn-γ and Il-4 were remarkably increased in Tcon cells by flow cytometry detection of single cell suspension from lymph-nodes (Fig. 1g and Supplementary Fig. 3d). In addition, the serum concentrations of $T_H1$/$T_H2$ cytokines (Ifn-γ, Il-2 for $T_H1$; Il-4, Il-5, Il6, Il-9, Il-10, and Il-13 for $T_H2$) (Supplementary Fig. 5a, b) and antibodies (IgA and IgG2a for $T_H1$; IgE and IgG1 for $T_H2$) (Supplementary Fig. 6a, b) were all elevated and, indicating a robust activation of $T_H1$/$T_H2$-immune responses. For $T_H17$ reaction, flow cytometry of single cell suspension from lymph-nodes revealed a two-fold and statistically significant increase of Il-17A in Tcon cells (Fig. 1h and Supplementary Fig. 3e), and serum concentrations of $T_H17$ antibodies (IgG3, IgG2b) were also elevated, although not reaching the statistical difference (Supplementary Fig. 6c). While the serum concentrations of $T_H17$ cytokines (Tnf-α, and GM-Csf) did not change obviously, the Il-17A level was actually reduced (Supplementary Fig. 5c), despite a robust

inflammatory storm was evident with significantly increased amount of various cytokines in *Foxp3*$^{cre}$;*Rbx1*$^{fl/fl}$ mice (Supplementary Fig. 5d). Thus, while the $T_H17$ reaction was activated modestly, an excessive $T_H1$/$T_H2$-dominant immune polarization was clearly shown in *Foxp3*$^{cre}$;*Rbx1*$^{fl/fl}$ mice.

**Rbx1 is required for the immune suppressive function of Treg cells.** To investigate whether the suppressive function of Rbx1-deficient Treg cells was impaired, we performed an in vivo suppression assay. The equal number of Treg cells from *wt* or *Rbx1*-deficient mice were injected into immune-deficient *Rag1*$^{−/−}$ mice, along with naive conventional CD4$^+$ T cells. If mice develop the colitis in 4 weeks, induced by co-injected naive conventional CD4$^+$ T cells, it would indicate that Treg cells from *Rbx1*-deficient mice have lost suppression function, and vice versa. We found that while *wt* Treg cells prevent the colitis, as evidenced by a normal colon morphology, the *Rbx1*-deficient

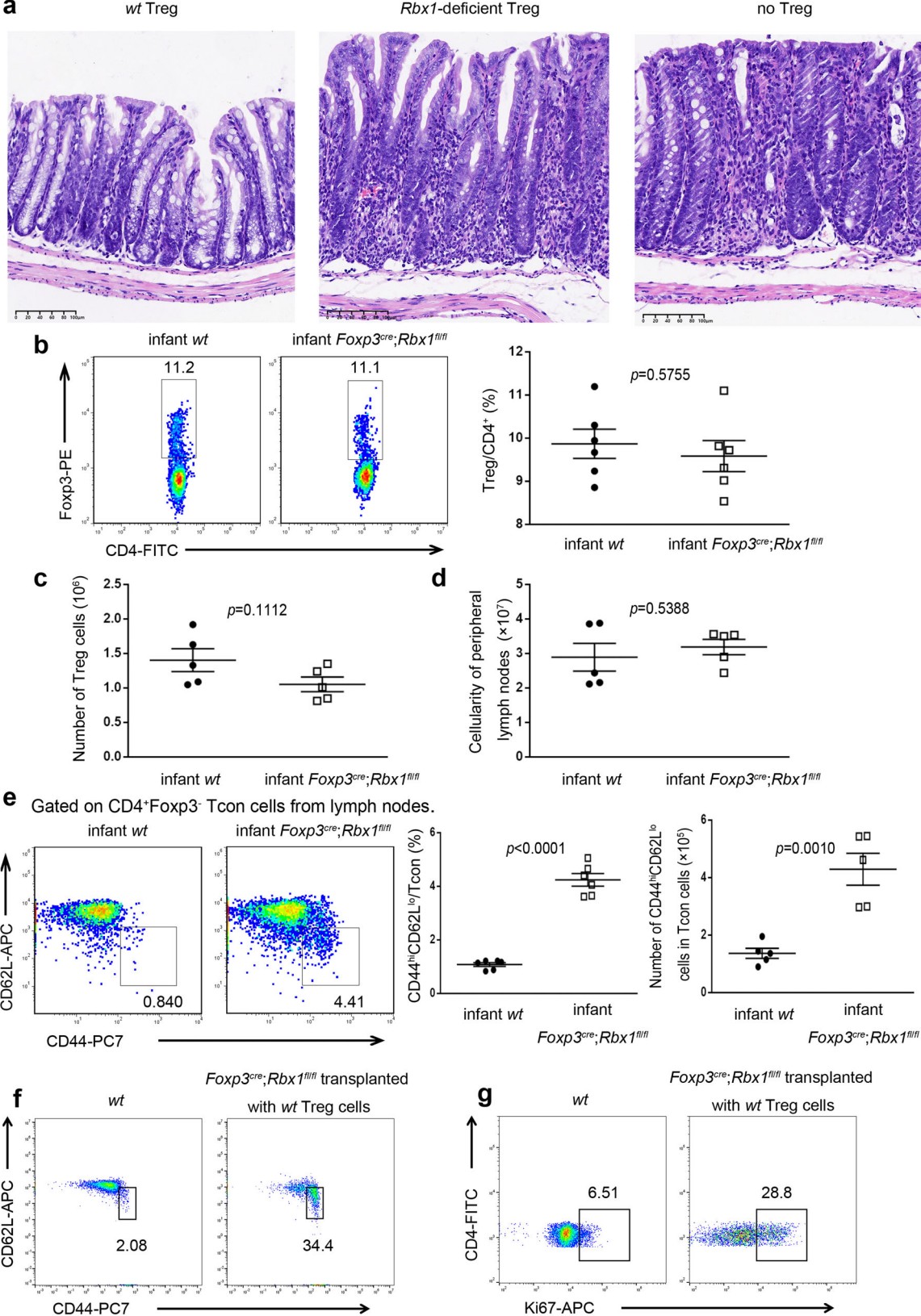

Treg cells failed to do so, and the mice developed very severe colitis, similar to the control mice without receiving Treg cell injection (Fig. 2a), demonstrating that the immune suppressive function of Treg cells was indeed impaired upon *Rbx1*-deletion.

To exclude the possibility that *Rbx1*-deficient Treg cells might die in the 4-week period during the in vivo suppression assay, we also used an alternative strategy to investigate the impairment of the immune suppressive function of *Rbx1*-deficient Treg cells at early age. Compared to the *wt* control mice, *Foxp3^cre^;Rbx1^fl/fl^* mice at the p8 stage have similar levels of Treg/CD4+ ratio (Fig. 2b), similar numbers of Treg cells, as well as total numbers of lymphocytes (Fig. 2c, d). However, these Treg *Rbx1*-deficient

**Fig. 2 Impaired suppressive function of Rbx1-deficient Treg cells. a** Representative images of distal colon after H&E staining (scale bar = 100 μm, 3 times each experiment was repeated independently with similar results). **b** Treg/CD4$^+$ ratios in peripheral lymph nodes from *wt* and *Foxp3$^{cre}$;Rbx1$^{fl/fl}$* infant mice (p8, n = 6 biologically independent samples from both male and female mice). **c** Treg cell numbers in peripheral lymph nodes from *wt* and *Foxp3$^{cre}$;Rbx1$^{fl/fl}$* infant mice (p8, n = 5 biologically independent samples from both male and female mice). **d** Total cell numbers in peripheral lymph nodes from *wt* and *Foxp3$^{cre}$;Rbx1$^{fl/fl}$* infant mice (p8, n = 5 biologically independent samples from both male and female mice). **e** Expression of CD44 and CD62L in Tcon cells from peripheral lymph nodes of *wt* and *Foxp3$^{cre}$;Rbx1$^{fl/fl}$* infant mice (p8, n = 5 biologically independent samples from both male and female mice, p < 0.0001); the right panel showed correspondent actual number of CD44$^{hi}$CD62L$^{lo}$ cells (p8, n = 5 biologically independent samples from both male and female mice, p = 0.0010). **f** Expression of CD44 and CD62L in Tcon cells from peripheral lymph nodes of *wt* and *Foxp3$^{cre}$;Rbx1$^{fl/fl}$* (transplanted with *wt* Treg cells) mice (p20, male). **g** Expression of Ki67 in Tcon cells from peripheral lymph nodes of *wt* and *Foxp3$^{cre}$;Rbx1$^{fl/fl}$* (transplanted with *wt* Treg cells) mice (p20, male). All error bars represent the SEM, data are presented as mean values +/− SEM. The p values were calculated by Mann–Whitney test. Source data are provided as a Source Data file.

infant mice still showed an over-activated immune system with a ~3–4-fold increase in proportion and cell number of effector/memory (CD44$^{hi}$CD62L$^{lo}$) T cells (Fig. 2e).

Finally, we performed the *wt* Treg infusion experiment and found that intraperitoneal injection of *wt* Treg cells to the infant *Foxp3$^{cre}$;Rbx1$^{fl/fl}$* mice cannot rescue the fatal inflammation (Fig. 2f, g). Specifically, compared to *wt* control, *Foxp3$^{cre}$;Rbx1$^{−/fl}$* mice (at the p8 stage) transplanted with *wt* Treg cells still developed immune over-activation (harvested at the p20 stage), revealed by elevated proportion of effector/memory T cells (CD44$^{hi}$CD62L$^{lo}$) among populations of Tcon cells (Fig. 2f), and increased proliferation rate of Tcon cells (Fig. 2g). The possible explanation is that despite Treg cells was reported to inhibit established inflammation in 5 weeks[30], the infused wt Treg cells did not have enough time window for functional rescue after inflammation had already started in the recipient infant *Foxp3$^{cre}$;Rbx1$^{fl/fl}$* mice. Taken together, all these three assays demonstrated that Rbx1 is essential and required for immune suppression function of Treg cells.

In moribund *Foxp3$^{cre}$;Rbx1$^{fl/fl}$* mice at p19–23 stage, Treg cellular compartment was reduced dramatically, as evidenced by reduced Treg cell ratios (Supplementary Fig. 7a) and actual numbers (Supplementary Fig. 7b), with reduced proliferation and increased apoptosis, revealed by reduced Ki67 expression (Supplementary Fig. 7c) and increased Caspase-3 activity (Supplementary Fig. 7d). Thus, the fatal inflammatory disorders in *Foxp3$^{cre}$;Rbx1$^{fl/fl}$* mice are attributable to both impaired suppressive function and decreased cellular compartment.

**Rbx1 is essential for the conversion of quiescent to effector subpopulations of Treg cells.** Single-cell RNA sequencing (scRNA-seq) was utilized to unbiasedly dissect the cellular heterogeneity of *wt* vs. *Rbx1*-deficient CD4$^+$YFP$^+$ Treg cells from the peripheral lymph nodes of inflammation-free female *Foxp3$^{cre/wt}$* and *Foxp3$^{cre/wt}$;Rbx1$^{fl/fl}$* mice (Note that the *Foxp3* gene is localized in the X-chromosome, and the female mice used here with two genotypes are chimeric, Supplementary Fig. 8). As revealed in the two-dimensional Uniform Manifold Approximation and Projection (UMAP) of the single-cell transcriptomics data, both *wt* and *Rbx1*-deficient Treg cells could be divided into several subpopulations (Fig. 3a and Supplementary Fig. 9). *Rbx1*-depletion increased the subpopulation in Cluster 0, but remarkably decreased the subpopulations on Clusters 3 and 4 without obviously affecting other subpopulations (Fig. 3a, b).

The top 10 genes and top 10 pathways in each of 14 clusters in Treg cells were shown (Supplementary Figs. 10, 12). Among them, the cells in Cluster 0 expressed high levels of genes characteristic of naive Treg cells, including *Sell* (*Cd62L*)[31] and *Bcl2*[32]; while Treg cell functional molecules, such as *Ctla4*[33], *Icos*[34], and *Cd44*[35], were highly expressed in Cluster 3 and 4 (Fig.3c). To confirm this finding, we performed the FACS-based profiling and found a ~3–4-fold reduction of CD44$^{hi}$CD62L$^{lo}$

cells in Treg population from peripheral lymph nodes in *Foxp3$^{cre}$;Rbx1$^{fl/fl}$* mice at age of ~20-day (Fig. 3d and Supplementary Fig. 11a). A similar reduction was also observed in infant *Foxp3$^{cre}$;Rbx1$^{fl/fl}$* mice at age of 8-day (Fig. 3e and Supplementary Fig. 11b), indicating that the reduction in Cluster 3/4 subpopulations upon Treg *Rbx1*-deletion is an early event.

The KEGG pathway enrichment analysis revealed that, compared to Cluster 0, T-cell receptor[36] and other Treg-related pathways were activated in Clusters 3 and 4 (Fig. 3f and Supplementary Fig. 12). Thus, Cluster 0 is a quiescent subpopulation, designated as the center population, whereas Clusters 3 and 4 are active Treg cell subpopulations, designated as effector populations[37]. The fact that the loss of *Rbx1* leads to the failure in the conversion of quiescent subpopulation to effector subpopulation supports the notion that Rbx1 is essential for such a conversion in the Treg cells.

**Disordered functional and regulatory network in *Rbx1*-deficent Treg cells.** Transcriptome analysis of *Rbx1*-deficient Treg cells from inflammation-free female *Foxp3$^{cre/wt}$;Rbx1$^{fl/fl}$* mice (Supplementary Fig. 8) revealed significant down-regulation of functional genes, including *Il10*[27], *Tgfb*[38], and *Cst7*[39] (suppressive cytokines); *Il2r*[40] and *Entpd1*[41,42] (regulators of immune cell metabolism); *Ctla4*[33], *Icos*[34], *Nrp1*[43], and *Lag3*[44] (inhibitor via dendritic cells); and *Fas*[45] and *Lgals1*[46] (apoptosis inducer). Impairment of migration-associated surface molecules was also observed in *Rbx1*-deficient Treg cells, including downregulation of *Cd44*, *Cxcr3*, *Cxcr5*, and *Cxcr6*[35] and upregulation of *Sell*[31] (Fig. 4a).

The KEGG pathway enrichment analysis revealed that Rbx1 depletion caused downregulation of transcriptome involved in many signaling pathways, including T-cell receptor, Th17 cell differentiation, chemokine and cytokine–cytokine receptor interaction, NOD-like receptor, TNF, FoxO, Toll-like receptor, Hippo, TGFβ, NF-κB, PI3K-AKT, and purine metabolism among few other virus associated pathways (Fig. 4b and Supplementary Fig. 13). The results suggest that loss of *Rbx1* results in changes in the expression of numerous key genes regulating the processes of inflammation, immunological responsiveness, proliferation/survival, and metabolisms, which are subjected to further experimental validation.

Considering Rbx1 is an E3 ubiquitin ligase that directly controls the abundance of substrate proteins, we utilized the mass spectrometry-based proteomics to profile the difference in protein levels between *wt* and *Rbx1*-deficient CD4$^+$YFP$^+$ Treg cells (three independent sets of combined samples) from inflammation-free female *Foxp3$^{cre/wt}$* and *Foxp3$^{cre/wt}$;Rbx1$^{fl/fl}$* mice (Supplementary Fig. 8). Among a total of 3236 proteins detected in any one of 6 paired samples, we identified 43 increased and 78 decreased proteins with a three-fold cut-off, and the numbers increased to 70 and 195, respectively, with a two-fold cut-off in Rbx1-deficient CD4$^+$YFP$^+$ Treg cells, as compared to the wt control. The KEGG pathway enrichment analysis of

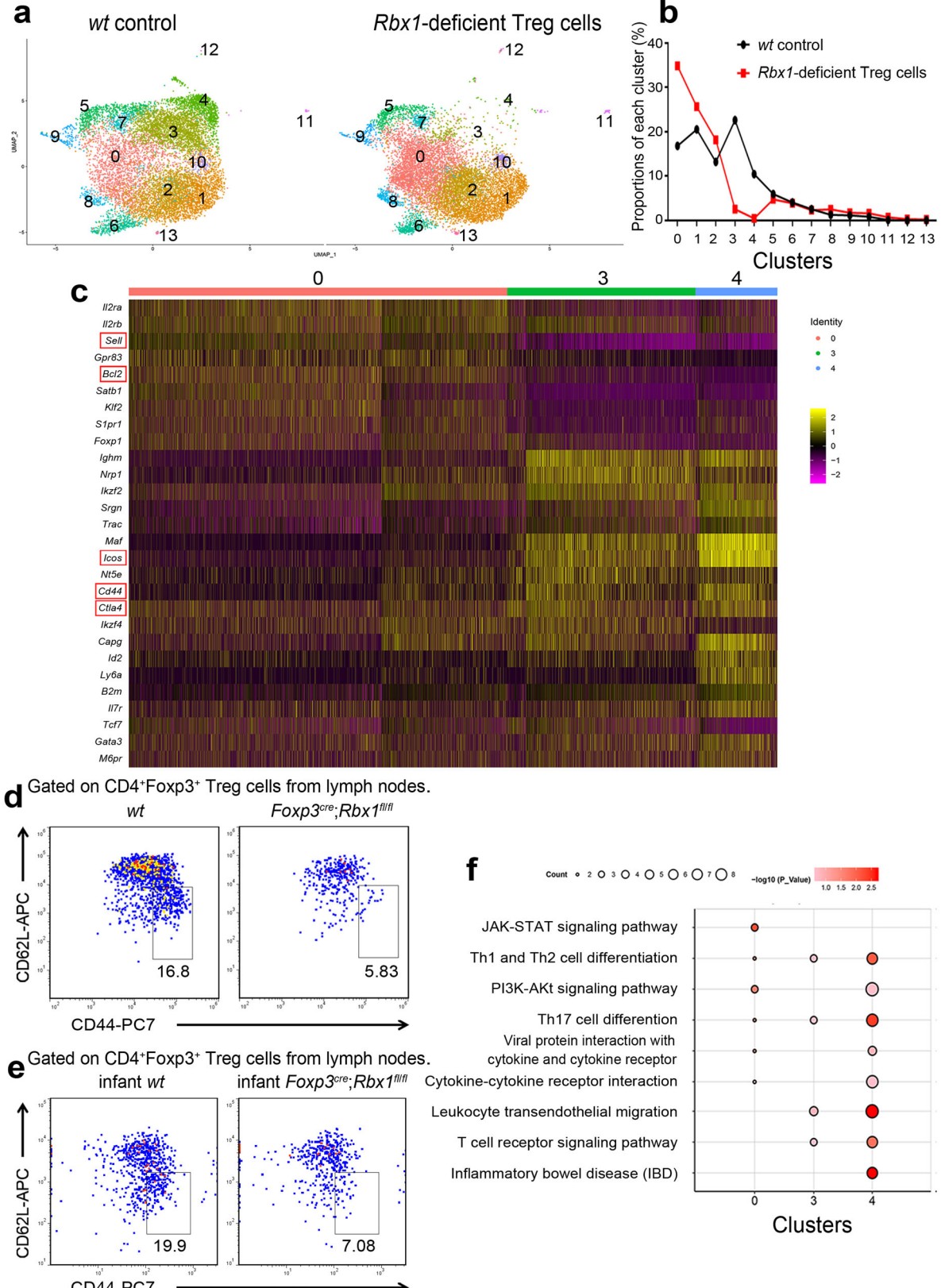

**Fig. 3 Single-cell transcriptomics of *wt* and *Rbx1*-deficient Treg cells. a** Two-dimensional UMAP visualization of single cell transcriptomics data in CD4[+]YFP[+] Treg cells from peripheral lymph nodes of female *Foxp3*[cre/wt] and *Foxp3*[cre/wt];*Rbx1*[fl/fl] mice (10 weeks old). **b** Proportions of each cluster in CD4[+]YFP[+] Treg cells in peripheral lymph nodes from female *Foxp3*[cre/wt] and *Foxp3*[cre/wt];*Rbx1*[fl/fl] mice (10 weeks old). **c** Heat map of selected marker genes in clusters 0, 3, and 4. Single cells were represented by vertical lines and different colors reflected the relative abundance of indicated genes. **d** Expression of CD44 and CD62L in Treg cells from peripheral lymph nodes of *wt* and *Foxp3*[cre];*Rbx1*[fl/fl] mice (p20, male). **e** Expression of CD44 and CD62L in Treg cells from peripheral lymph nodes of *wt* and *Foxp3*[cre];*Rbx1*[fl/fl] infant mice (p8, male). **f** Selected pathways of clusters 0, 3, and 4 in the KEGG analysis. Source data are provided as a Source Data file.

two-fold cut-off revealed that Rbx1 depletion caused an obvious change of many pathways (Supplementary Fig. 14), including accumulation of few proteins associated with the splicesome or complement and coagulation cascade, but significant reduction of proteins involving ER processing, and of proteins controlling the various metabolic pathways, including TCA cycle, carbon and purine metabolisms, along with other metabolic pathways (Fig. 4c, d). We marked (with arrows) the genes with the consistent changes at both mRNA and protein levels, and these observed changes at the protein levels are likely due to altered gene transcription (Fig. 4c, d), which is again subjected to further experimental validation.

**Specific spectrum of Rbx1 substrates in Treg cells**. To define candidates as possible direct substrates of Rbx1 ubiquitin E3 ligase in Treg cells, the proteomic and transcriptomic data were compared directly. We reasoned that the substrate candidates should be those with the increase at the levels of protein, but not of mRNA, in Rbx1-deficient Treg cells. A total of 57 such candidates were identified (Fig. 5a). While Acly[47], Bcl2l11/Bim[48], Ets1[49], Nfkbib/IκB-β[50], and Zhx2[51] were previously reported as Rbx1 substrates, suggesting the reliability of our methodology, several others in the list are likely the novel substrates for future validation.

Interestingly, many common substrates of Rbx1/CRLs1-4[1,6] were not detected in Rbx1 deficient Treg cells, and few detected ones, like Cdkn1b/p27[7] and Foxo1[52], were not accumulated (Fig. 5b). Thus, it appears that only a small subset of Rbx1 substrates are selectively expressed, and specifically subjected to Rbx1 modulation in Treg cells to maintain their proper functions.

**The involvement of Bim in the phenotypes of $Foxp3^{cre};Rbx1^{fl/fl}$ mice**. In order to validate these putative Rbx1 substrates in Treg cells, we searched all 57 proteins for antibodies commercially available for FACS-based quantification, given the difficulty in harvesting sufficient Treg cells for traditional Western blotting analysis. Unfortunately, only available antibody is for Bim. We, therefore, performed the FACS analysis, as described[53], to quantify the levels of Bim, and confirmed that Bim is indeed accumulated up to three-fold in Rbx1-depleted Treg cells (Fig. 6a, b). It is noting worthy that our single cell RNA sequencing data revealed that $Bim$ mRNA is widely expressed in all subpopulations of Treg cells (Supplementary Fig. 15a).

Given Bim is a pro-apoptotic protein[54], and also involved in Treg cells regulation[55], Bim accumulation could mediate at least in part some phenotypes of $Rbx1$-deficient Treg cells. We tested this working hypothesis via a Bim rescue experiment by generating and characterizing conditional knockout mice with inactivation of both $Rbx1$ and $Bim$ in Treg cells driven by $Foxp3^{YFP-cre}$ ($Foxp3^{cre}$)[27,28] (designated as $Foxp3^{cre};Rbx1^{fl/fl};Bim^{fl/fl}$ mice), which have more than 10-fold reduction of Bim levels in Treg cells (Supplementary Fig. 15b). The $Foxp3^{cre};Rbx1^{fl/fl};Bim^{fl/fl}$ mice also developed an early-onset fetal inflammatory disorder with very short lifespan, similar to the $Foxp3^{cre};Rbx1^{fl/fl}$ mice (Fig. 6c, d), along with decreased $CD4^+/CD8^+$ T-cell ratio (Supplementary Fig. 15c) and significantly increased proportion of effector/memory T cells ($CD44^{hi}CD62L^{lo}$) among populations of Tcon cells (Fig. 6e and Supplementary Fig. 15d). Notably, the above inflammation markers, however, were milder in double cKO $Foxp3^{cre};Rbx1^{fl/fl};Bim^{fl/fl}$ mice than those in $Foxp3^{cre};Rbx1^{fl/fl}$ mice, with the statistically significant $p$ values (Fig. 6e, f), suggesting the inflammation and immune over-activation induced by $Rbx1$-deficiency in Treg cells were moderately attenuated by $Bim$-deletion. Moreover, Treg ratio among $CD4^+$-T cells in $Foxp3^{cre};Rbx1^{fl/fl};Bim^{fl/fl}$ mice remained significantly

reduced, as compared to $wt$ control (Fig. 6g and Supplementary Fig. 15e), despite slightly higher than that in $Foxp3^{cre};Rbx1^{fl/fl}$ mice (Fig. 6h). However, the absolute numbers of Treg cells in $Foxp3^{cre};Rbx1^{fl/fl};Bim^{fl/fl}$ mice were comparable to the $wt$ control mice (Supplementary Fig. 15f), while it was significantly reduced in $Foxp3^{cre};Rbx1^{fl/fl}$ mice (Supplementary Fig. 7b). Notably, Treg cells with double deletion of $Rbx1$ and $Bim$ still suffer from elevated apoptosis (Supplementary Fig. 15g), in a level similar to Rbx1-deficient Treg cells (Supplementary Fig. 7d). Taken together, $Bim$ deletion in Treg cells attenuated the severe inflammation phenotype, although it failed to rescue the survival of $Foxp3^{cre};Rbx1^{fl/fl}$ mice, indicating Bim is partially involved in the function of Rbx1 in Treg cells.

**Later-onset fatal inflammation in $Foxp3^{cre};Ube2m^{fl/fl}$ mice with impaired suppressive function of Treg cells**. It has been well-established that the E3 ligase activity of the CRLs requires neddylation of the scaffold protein Cullin[56,57]. Specifically, neddylation E2 Ube2m couples with Rbx1 (neddylation co-E3) selectively promotes neddylation of Cullins 1–4, which are complexed with Rbx1 to constitute active CRLs 1–4 ubiquitin E3; whereas neddylation E2 Ube2f couples with Sag/Rbx2 E3 to promote neddylation of Cullin 5, which is then complex with Sag/Rbx2 to constitute active CRL5[2]. We, therefore, asked whether the pivotal role of Rbx1 in Treg cells is dependent on Ube2m, whereas Ube2f is non-essential for Treg cells, given the non-essentiality of Sag/Rbx2 (Supplementary Fig. 1). Indeed, despite Ube2f is expressed in $wt$ Treg cells (Supplementary Fig. 16a), $Foxp3^{cre};Ube2f^{fl/fl}$ mice with $Ube2f$ deletion in Treg cells are viable and develop normally without any abnormality, as evidenced by a normal level of an activation marker of $CD4^+Foxp3^-$ T cells (Supplementary Fig. 16b), a normal rate of Tcon cell proliferation (Supplementary Fig. 16c), and a normal Treg/$CD4^+$ ratio (Supplementary Fig. 16d), indicating the Ube2f-Rbx2-CRL5 axis is not essential for the functionality and survival of Treg cells at the steady status.

On the other hand, like $Foxp3^{cre};Rbx1^{fl/fl}$ mice, $Foxp3^{cre};Ube2m^{fl/fl}$ mice with $Ube2m$ deletion in Treg cells (Supplementary Fig. 17a), displayed abnormal appearances such as collapsed ears and festered skin, but at the much later stage (Fig. 7a). The growth of mice was remarkably retarded (Fig. 7b), and about 50% of the mice dies at age of 150 days and a 100% death rate within one year (Fig. 7c). The swollen spleens and lymph nodes were apparent with increased cellularity (Fig. 7d and Supplementary Fig. 17b) with moderate lymphocyte infiltration into multiple organs, including skin, lung, stomach, liver, kidney, and colon (Supplementary Fig. 17c). Furthermore, the $Foxp3^{cre};Ube2m^{fl/fl}$ mice also showed obvious $T_H1/T_H2$-polarized immune over-activation of multiple immune cells (Fig. 7e–g and Supplementary Figs. 18–21), but in a much lower degree of inflammation than that in $Foxp3^{cre};Rbx1^{fl/fl}$ mice (Fig. 1 and Supplementary Figs. 2–6).

Interestingly, unlike $Foxp3^{cre};Rbx1^{fl/fl}$ mice, $Foxp3^{cre};Ube2m^{fl/fl}$ mice had increased Treg/$CD4^+$-T ratio and increased number of Treg cells (Fig. 8a) in lymph nodes, without obvious alterations in the levels of Ki67 and active Caspase-3 in Treg cells (Fig. 8b, c). It has been shown that in response to inflammation insult, the immuno-suppressive Treg cells usually proliferate to inhibit the inflammation and restore the immune homeostasis of the body[58,59]. The ratio (Treg/$CD4^+$) and number of Treg cells are, therefore, elevated in vivo under inflammation, if the proliferation and survival of Treg cells are not inhibited, which is the case for $Ube2m$-deficient Treg cells, but not for $Rbx1$-deficient Treg cells. Nevertheless, the fact that $Foxp3^{cre};Ube2m^{fl/fl}$ mice still developed over-reactive immune response (Fig. 7) strongly suggested

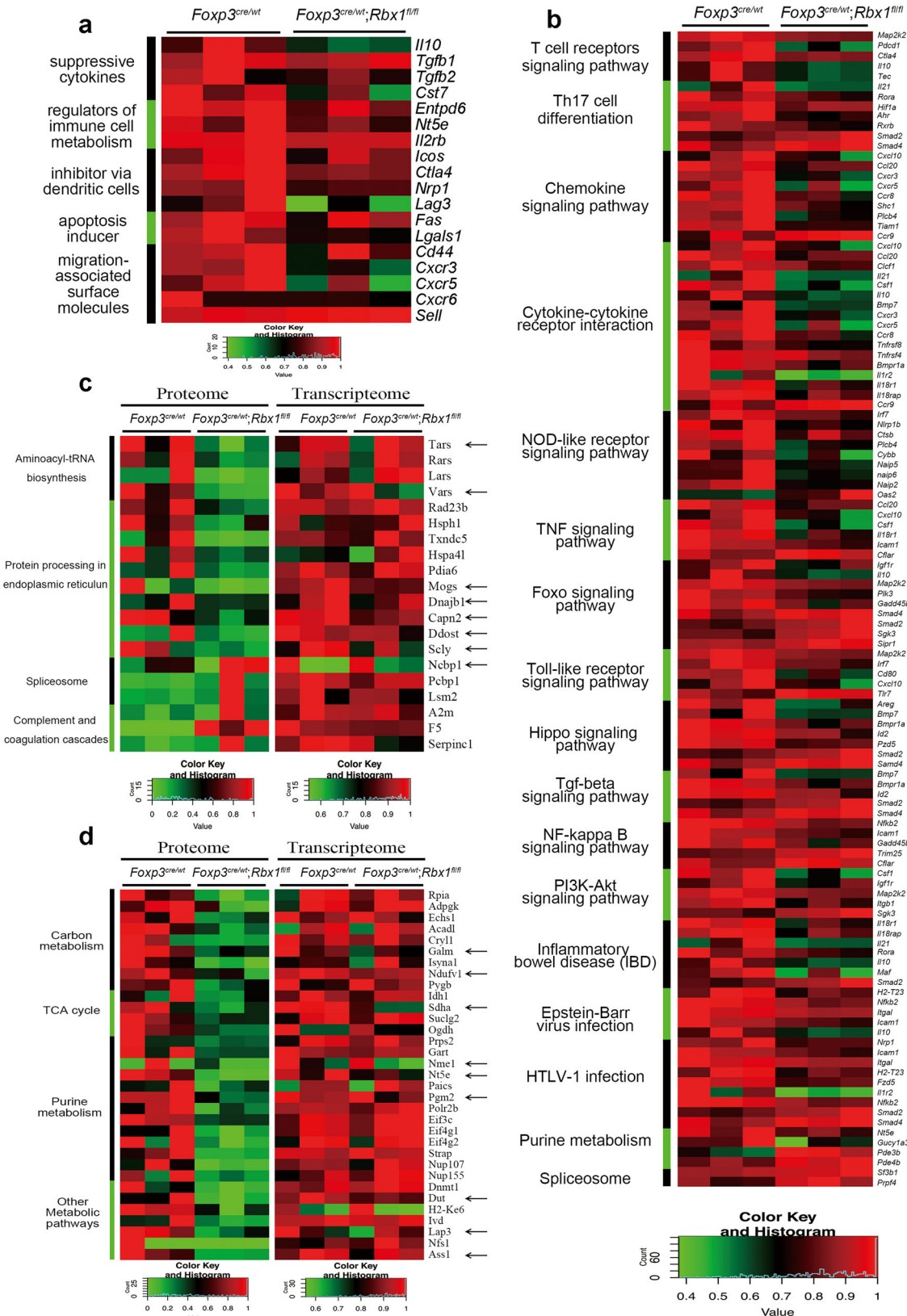

the impairment of suppressive function of Ube2m-deficient Treg cells. Indeed, the in vivo suppression assay using immune-deficient $Rag1^{-/-}$ mice clearly showed that very severe colitis, induced by injection of naive CD4+-T cells, cannot be inhibited by Ube2m-deficient Treg cells (Fig. 8d). However, unlike that of $Foxp3^{cre};Rbx1^{fl/fl}$ mice, the inflammatory immune phenotype of

$Foxp3^{cre};Ube2m^{fl/fl}$ mice can be largely rescued by intraperitoneal injection of wt Treg cells, as evidenced by a largely normal proportion of both effector/memory T cells (CD44hiCD62Llo) among populations of Tcon cells (Fig. 8e) and proliferation rate of Tcon cells (Fig. 8f), suggesting that impaired immune suppressive function in $Foxp3^{cre};Ube2m^{fl/fl}$ mice are relatively

**Fig. 4 Rbx1-dependent transcriptome and proteome in Treg cells. a** Differentially expressed genes related to Treg function in CD4+YFP+ Treg cells from female *Foxp3cre/wt* and *Foxp3cre/wt;Rbx1fl/fl* mice (8–10 weeks old), determined by transcriptional profiling. The known functional molecules of Treg cells were selected, and those with fold change greater than 1.27 were displayed in the heat-map. **b** Comparison of expression of genes associated with indicated pathways in CD4+YFP+ Treg cells derived from female *Foxp3cre/wt* and *Foxp3cre/wt;Rbx1fl/fl* mice (8–10 weeks old). KEGG analysis was performed and genes with changes greater than 2 times and *p* value less than 0.05 (*p* < 0.05) were selected; biologically significant and interesting pathways were selected and displayed in the heat-map. **c, d** Comparison of the levels of proteins vs. mRNAs in genes associated with indicated pathways in CD4+YFP+ Treg cells from female *Foxp3cre/wt* and *Foxp3cre/wt;Rbx1fl/fl* mice (8–10 weeks old). The mRNAs with greater than 1.3-fold changes, which are consistent with protein level changes, were marked by the arrows. KEGG analysis was performed and proteins with changes greater than two times and *p* value less than 0.05 (*p* < 0.05) were selected; biologically significant and interesting pathways were selected and displayed in the heat-map: **c** various pathways, **d** metabolism-related pathways.

minor, as compared to *Foxp3cre*;Rbx1fl/fl mice and can be rescued by *wt* Treg cells in this infusion experiment.

**Comparison of signaling pathways regulated by Rbx1 and Ube2m in Treg cells**. The phenotypes between *Foxp3cre*;Rbx1fl/fl and *Foxp3cre*;Ube2mfl/fl mice clearly demonstrate the similarities and differences between Rbx1 and Ube2m in the maintenance of the functionality of Treg cells. To dissect them at the molecular levels, we performed the genome-wide transcriptome analyses of CD4+YFP+ Treg cells derived from female *Foxp3cre/wt;Ube2mfl/fl* and *Foxp3cre/wt;Rbx1fl/fl* mice (Supplementary Fig. 8). The results showed the similar changes in the Treg functional genes (Supplementary Fig. 22), and multiple signaling pathways involved in inflammation responses, but generally to a lesser extent in *Ube2m*-deficient Treg cells (Supplementary Fig. 23). An unbiased cluster analysis of the transcriptomes between *Rbx1*-deficient and *Ube2m*-deficient Treg cells revealed a good overall correction, but two sets of data did have unique differences in several groups of genes and pathways, particularly seen in *Rbx1*-deficient Treg cells (Fig. 9a and Supplementary Figs. 24, 25), demonstrating that CRLs and neddylation did have common as well as unique mechanisms in the regulation of Treg cell functions.

Taken together, both the phenotypic comparison and unbiased cluster analyses of the transcriptome support the notion that Rbx1 plays Ube2m-dependent and -independent roles in Treg cells (Fig. 9b).

**Discussion**

In this study, we systematically investigated the role of CRLs and neddylation in the maintenance of fitness of Treg cells, using four Treg selective KO mouse models. While the Ube2f/Sag axis plays a minimal, if any, role, the Ube2m/Rbx1 axis is indeed essential for the homeostasis and survival of Treg cells, although all four genes are widely expressed in multiple mouse tissues[60] and in all subpopulations in Treg cells (Supplementary Fig. 26). The finding that mice of *Rbx1* Treg deletion suffer from much more severe autoimmune disorders with much earlier fatality than those of *Ube2m* Treg deletion is unexpected, since Ube2m is an upstream E2, whereas Rbx1 is a downstream co-E3 in the neddylation cascade. Thus, our in vivo phenotypical study via targeting Treg cells clearly demonstrated that Rbx1/CRLs1–4 have additional functions beyond neddylation activation (Fig. 9b), as generally accepted, which was mainly based upon the in vitro biochemical and cell biology studies[2]. Furthermore, severe autoimmune phenotype of *Foxp3cre*;Rbx1fl/fl mice is reminiscent of mice devoid of Treg cells[23] or deficient in *Foxp3*[29]. Very few genes, upon Treg deletion in mice, have shown a severe autoimmune phenotype, similar to that of the Treg-deficiency, including *Foxo1*[58], *Raptor*[59], *Lkb1*[53], *Uqcrsf1*[61], and *Pggt1b*[62]. Thus, *Rbx1*, encoding a dual E3s for ubiquitylation and neddylation, is a new addition to

this short list as one of the most critical genes, absolutely required for the functionality of Treg cells.

Several lines of evidence suggest that severe autoimmune/inflammatory phenotype of *Foxp3cre*;Rbx1fl/fl mice is not the simple consequence of reduced proliferation and/or enhanced apoptosis, rather it is also attributable to the impaired immune suppression of *Rbx1*-deficent Treg cells. First, in an in vivo suppression assay, *Rbx1*-deficient Treg cells failed to inhibit the colitis induced by naïve CD4+ T cells in immune-deficient *Rag1−/−* mice (Fig. 2a); Second, the 8-day-old infant *Foxp3cre*;Rbx1fl/fl mice with normal ratio of Treg/CD4+-T cells and normal number of Treg cells already showed the immune activation (Fig. 2b–e); third, infusion of wt Treg cells into *Foxp3cre*;Rbx1fl/fl mice failed to rescue inflammation phenotype within the testing time window (Fig. 2f, g); and forth, simultaneous deletion of *Bim* and *Rbx1* failed to extend mouse life-span and rescued the severe inflammation phenotypes (Fig. 6).

So far, the only reported study of CRL regulation of Treg cells was the Treg deletion of *Vhl*, a substrate receptor of CRL2 for targeted degradation of Hif1α[63]. In that study, the authors showed that mice with *Vhl*-deficient Treg cells also developed severe inflammatory disorders with a 50% of death rate at the age of 8 weeks[16]. Mechanistically, the production by *Vhl*-deficient Treg cells of a large amount of Ifn-γ due to transactivation by accumulated Hif1α is responsible for immune-disorder phenotype, which was completely rescued by simultaneous Ifn-γ deficiency in Treg cells[16]. In contract, the production of Ifn-γ by *Rbx1*-deficient Treg cells is comparable with that of *wt* controls (Supplementary Fig. 27), although other T cells (non-Treg cells) produced more Ifn-γ (Fig. 1g and Supplementary Fig. 3d), indicating a different mechanism of action between *Vhl* and *Rbx1* in functional regulation of Treg cells.

Our Microwell-based scRNA-seq revealed unbiasedly that Treg cells were divided into several subpopulations, and the quiescent population was significantly increased, whereas the effector subpopulations were significantly reduced upon Rbx1 depletion (Fig. 3), highly suggesting a failure in the conversion of the quiescent subpopulations into effector ones, rather than the death of these effector cells. Mechanistically, although we identified several altered signaling pathways (Fig. 3f), how exactly Rbx1 depletion failed to trigger such a conversion is currently unknown. Given the major role of Rbx1 as a dual E3 in both neddylation and ubiquitylation, it is very likely that Rbx1 depletion would result in the abnormal levels or activities of transcription factors and/or signaling molecules directly (via substrate accumulation or neddylation modification) or indirectly (altered transcriptions due to direct effect). Future studies are geared to elucidate the underlying mechanism. On the other hand, the fact that Rbx1-deficiency has no effect on other subpopulations of Treg cells provides an ideal opportunity of Rbx1 targeting for precise manipulation of Treg cell subpopulations.

It is worth noting that droplet-based and plate-based strategies were previously used for the scRNA-seq transcriptomics analysis

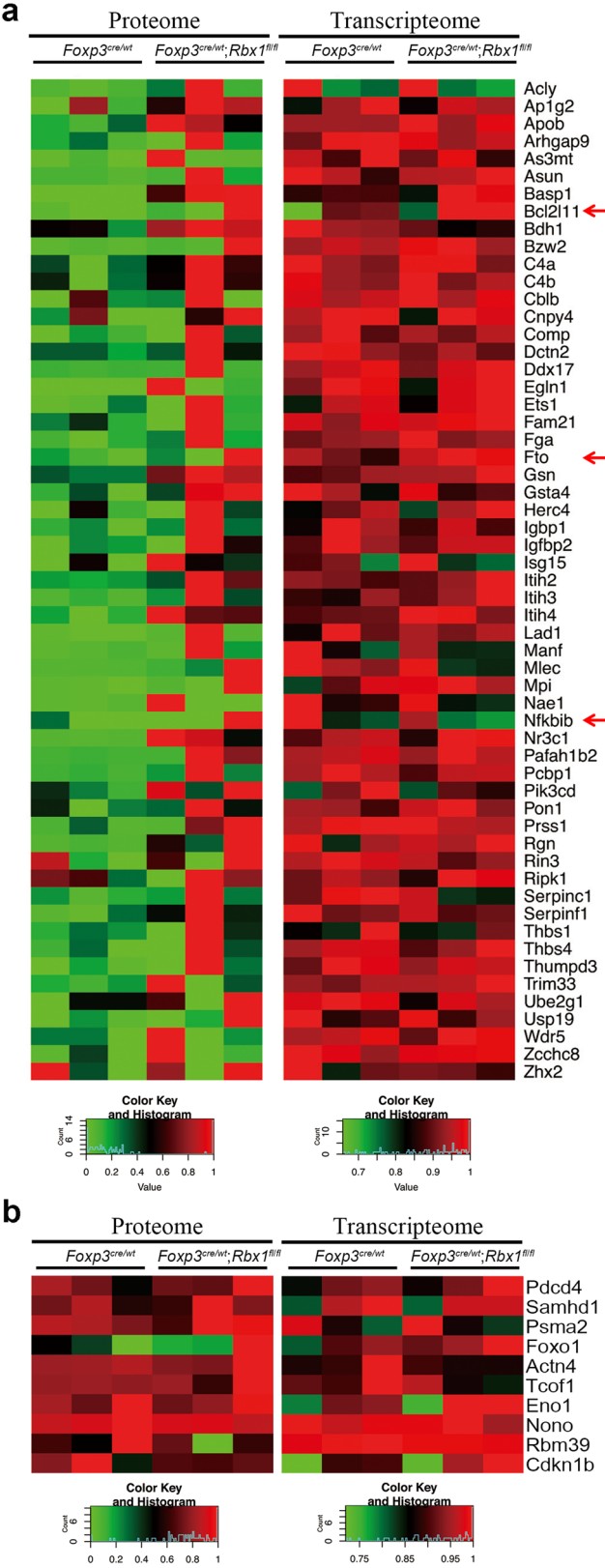

**a**

Proteome | Transcripteome
*Foxp3*cre/wt | *Foxp3*cre/wt;*Rbx1*fl/fl | *Foxp3*cre/wt | *Foxp3*cre/wt;*Rbx1*fl/fl

Acly
Ap1g2
Apob
Arhgap9
As3mt
Asun
Basp1
Bcl2l11 ←
Bdh1
Bzw2
C4a
C4b
Cblb
Cnpy4
Comp
Dctn2
Ddx17
Egln1
Ets1
Fam21
Fga
Fto ←
Gsn
Gsta4
Herc4
Igbp1
Igfbp2
Isg15
Itih2
Itih3
Itih4
Lad1
Manf
Mlec
Mpi
Nae1
Nfkbib ←
Nr3c1
Pafah1b2
Pcbp1
Pik3cd
Pon1
Prss1
Rgn
Rin3
Ripk1
Serpinc1
Serpinf1
Thbs1
Thbs4
Thumpd3
Trim33
Ube2g1
Usp19
Wdr5
Zcchc8
Zhx2

**b**

Proteome | Transcripteome
*Foxp3*cre/wt | *Foxp3*cre/wt;*Rbx1*fl/fl | *Foxp3*cre/wt | *Foxp3*cre/wt;*Rbx1*fl/fl

Pdcd4
Samhd1
Psma2
Foxo1
Actn4
Tcof1
Eno1
Nono
Rbm39
Cdkn1b

**Fig. 5 Comparison of the levels of proteins vs. mRNAs of indicated genes in CD4$^+$YFP$^+$ Treg cells from *Foxp3*$^{cre/wt}$ and *Foxp3*$^{cre/wt}$;*Rbx1*$^{fl/fl}$ mice (8–10 weeks old). a** Comparison of the levels of proteins vs. mRNAs in indicated genes in CD4$^+$YFP$^+$ Treg cells from female *Foxp3*$^{cre/wt}$ and *Foxp3*$^{cre/wt}$;*Rbx1*$^{fl/fl}$ mice (8–10 weeks old). Those with increased levels of protein but not of mRNA were considered as candidates of the direct degradation substrates of Rbx1 E3 ligase in Treg cells. **b** Some common substrates of Rbx1 are not accumulated in Rbx1-deficient Treg cells.

provided informative knowledge as to how Rbx1 regulates Treg subpopulations. Furthermore, this study is the first to provide detailed informative heat-map of top 10 characteristic genes in each of 14 clusters via scRNA sequencing, along with associated pathways (Supplementary Figs. 10, 12).

Both proteomics and transcriptome approaches were employed to analyze the effect of Rbx1-deficiency on Treg cells at the molecular levels. We noticed that less than half of protein changes seen in the proteomic analysis matched with the mRNA changes by the transcriptomic analysis (Fig. 4c, d). While this is slightly lower than the general correlation of 50% between proteome and transcriptome[64], the correlation between proteome and transcriptome is known to be distinct in different tissues or cell types[65], including in human Treg cells, as reported recently[66], likely due to the low abundance of Treg population in vivo with limited detection in our study of only ~3200 proteins in any given 6 samples by Mass Spectrum analysis. The comparison of proteome and transcriptome of Treg cells should provide some cues for future investigation of detailed mechanisms of Treg cell regulation by CRLs and neddylation.

The combination of proteomic and transcriptomic data revealed several Treg-specific Rbx1 degradation candidates, including Acly[47], Bcl2l11/Bim[48], Ets1[49], Nfkbib/IκB-β[50], and Zhx2[51]. Among the putative substrates of Rbx1 in Treg cells, Nfkbib/IκB-β is the inhibitor of NF-κB pathway, Fto is the demethylation enzyme of m$^6$A mRNA methylation, Ets1 is a transcription repressor, whereas Bcl2l11/Bim is a pro-apoptotic protein. The NF-κB pathway[28,67], m$^6$A mRNA methylation[68], and Bcl2l11/Bim[55] are all known to regulate Treg cell. Our *Bim* rescue experiment showed a partially role of Bim in Treg Rbx1-deficient phenotypes (Fig. 6), suggesting the severe inflammation phenotype is likely caused by the combination of multiple substrates. Future studies should be directed to characterize the substrates, particularly those involved in the regulation of global gene expressions (such as Fto and Ets1) in the functional regulation of Treg cells upon *Rbx1* deletion.

While Treg Ube2m-deficiency also causes obvious inflammation disorders (Fig. 7), the phenotype severity is much lesser than that seen in Treg Rbx1-deficient mice (Fig. 1). Mechanistically, although both Rbx1 and Ube2m controls the expression of genes regulating several inflammation-related pathways (Supplementary Fig. 23), the downregulation of *Egr2*[69], *Ccr8*[70] and *Tigit*[71] is rather unique to Rbx1-deficient Treg cells (Fig. 9a and Supplementary Fig. 24). Altered expression of these genes, which has been previously shown to regulate Treg cell function, would likely be responsible for the more severe phenotype seen in *Foxp3*$^{cre}$;*Rbx1*$^{fl/fl}$ mice.

In summary, we used in this study three comprehensive approaches, namely single cell sequencing, whole transcriptome profiling, and mass-spect based proteomics, in an attempt to define the mechanism of Rbx1 action in Treg cells. Given the broad role of the neddylation-CRLs in fundamental biological processes, our study yielded a wealth of information on many altered signaling pathways and individual gene/protein upon *Rbx1* Treg deletion. We reason that this is the first overall

of Treg cells[37]. The subpopulations of Treg cells revealed by our study were largely in agreement with the previous report[37] with exception that the center and effector populations were combined as a single population in that report[37]. The other minor differences are likely derived from different platforms. Nevertheless, a direct comparison between *wt* and *Rbx1*-null Treg cells indeed

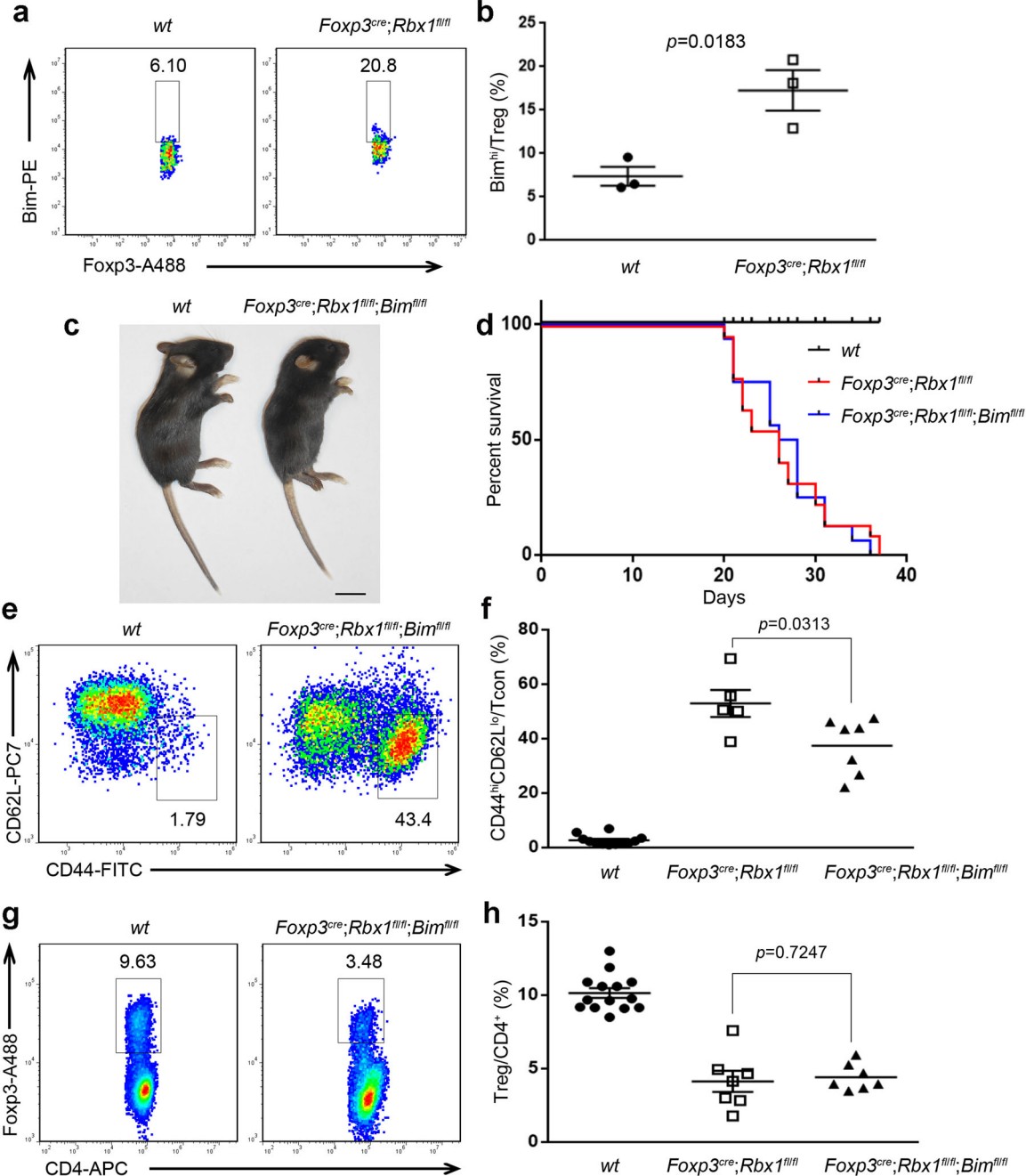

**Fig. 6 Involvement of Bim in the phenotypes of *Foxp3^cre;Rbx1^fl/fl* mice. a** Bim in Treg cells from peripheral lymph nodes of *wt* and *Foxp3^cre;Rbx1^fl/fl* mice (p19, male). **b** The proportion of Bim^hi cells among Treg cells in peripheral lymph nodes from *wt* and *Foxp3^cre;Rbx1^fl/fl* mice (p19–21, n = 3 biologically independent samples from both male and female mice). **c** Representative images of *wt* and *Foxp3^cre;Rbx1^fl/fl;Bim^fl/fl* mice (p23, male, scale bar = 1 cm). **d** Survival curve of *wt*, *Foxp3^cre;Rbx1^fl/fl* and *Foxp3^cre;Rbx1^fl/fl;Bim^fl/fl* mice (n = 16 animals for *Foxp3^cre;Rbx1^fl/fl;Bim^fl/fl* mice from both male and female mice). **e** Expression of CD44 and CD62L in Tcon cells from peripheral lymph nodes of *wt* and *Foxp3^cre;Rbx1^fl/fl;Bim^fl/fl* mice (p20, male). **f** CD44^hiCD62L^lo/Tcon ratios in peripheral lymph nodes from *wt*, *Foxp3^cre;Rbx1^fl/fl* and *Foxp3^cre;Rbx1^fl/fl;Bim^fl/fl* mice (p19–23, n = 11, 4, or 7 biologically independent samples respectively from both male and female mice). **g** The proportion of Treg cells among CD4^+-T cells from peripheral lymph nodes of *wt* and *Foxp3^cre;Rbx1^fl/fl;Bim^fl/fl* mice (p20, male). **h** Treg/CD4^+ ratios in peripheral lymph nodes from *wt*, *Foxp3^cre;Rbx1^fl/fl* and *Foxp3^cre;Rbx1^fl/fl;Bim^fl/fl* mice (p19–23, n = 13, 6, or 7 biologically independent samples respectively from both male and female mice). All error bars represent the SEM, data are presented as mean values +/− SEM. The *p* values were calculated by Mann–Whitney test. Source data are provided as a Source Data file.

phenotypical study of elucidating the role of neddylation enzymes, particularly Rbx1/Ube2m, in Treg cells, which sets the foundation for future detailed studies to pinpoint special role of each interesting pathways and individual genes. Given that inactivation of either neddylation E2/Ube2m, or E3/Rbx1 caused

suppression of Treg function, MLN4924, a small molecule inhibitor of neddylation, currently in several Phase II clinical trials for anticancer application[3,6] may have a new application in the treatment of diseases associated with over-activation of Treg cells, which is certainly worth investigating experimentally.

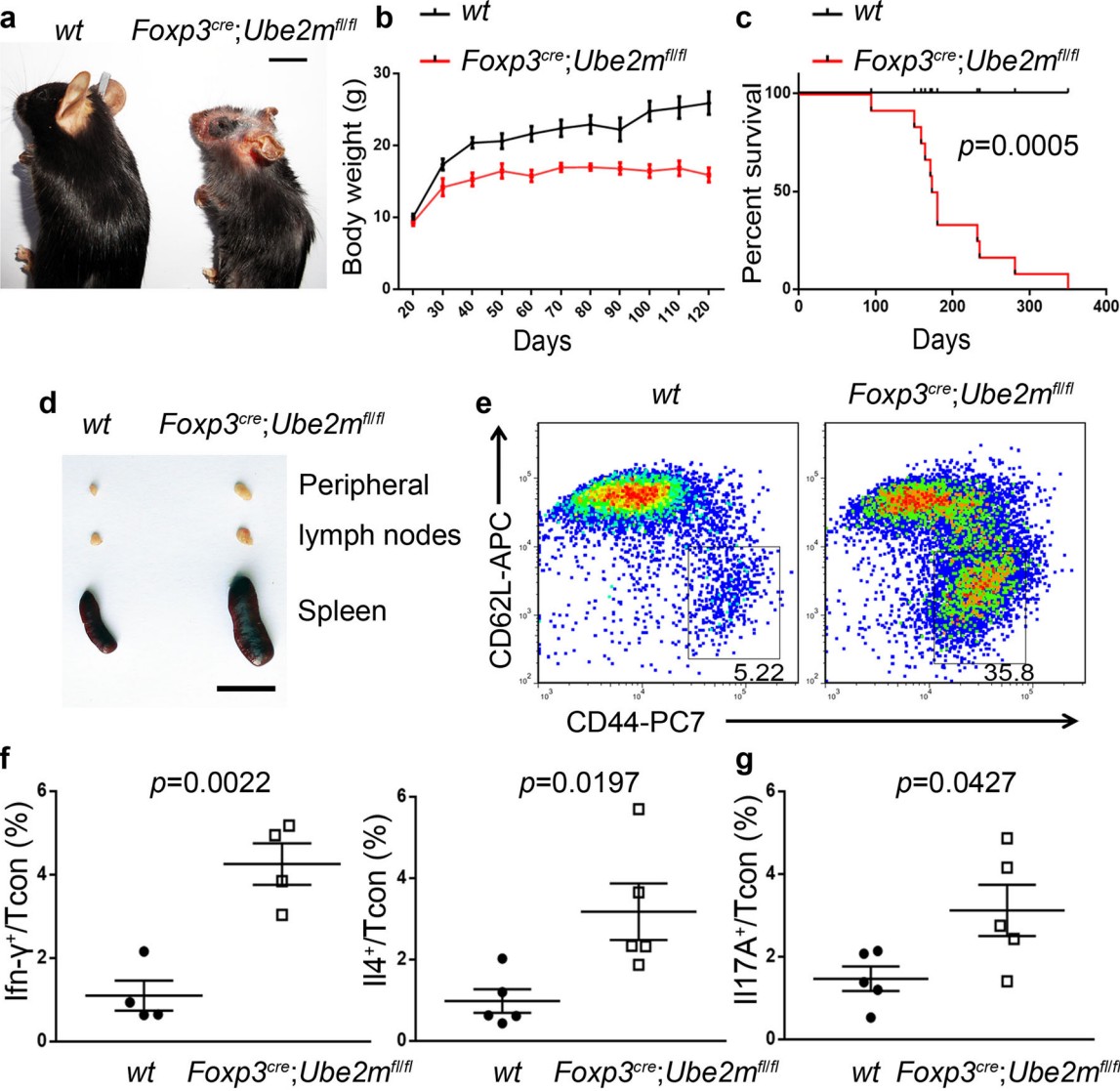

**Fig. 7 Fetal, but relative weaker, inflammatory disorder in _Foxp3cre;Ube2mfl/fl_ mice. a** Representative images of _wt_ and _Foxp3cre;Ube2mfl/fl_ mice (16 weeks old, male, scale bar = 1 cm). **b** Gross body weight of _wt_ and _Foxp3cre;Ube2mfl/fl_ mice ($n = 20$ animals, for gender information see "source data"). **c** Survival curve of _wt_ and _Foxp3cre;Ube2mfl/fl_ mice ($n = 12$ animals from both male and female mice; $p < 0.0001$). **d** Representative images of the peripheral lymph nodes and spleen from _wt_ and _Foxp3cre;Ube2mfl/fl_ mice (16 weeks old, male, scale bar = 1 cm). **e** Expression of CD44 and CD62L in Tcon cells from peripheral lymph nodes of _wt_ and _Foxp3cre;Ube2mfl/fl_ mice (16 weeks old, male). **f** The levels of Ifn-γ and Il-4 in Tcon cells from peripheral lymph nodes of _wt_ and _Foxp3cre;Ube2mfl/fl_ mice (16 weeks old, $n = 4$ or 5 biologically independent samples respectively from both male and female mice). **g** The levels of Il-17A in Tcon cells from peripheral lymph nodes of _wt_ and _Foxp3cre;Ube2mfl/fl_ mice (16 weeks old, $n = 5$ biologically independent samples from both male and female mice). All error bars represent the SEM, data are presented as mean values +/− SEM. The $p$ values were calculated by Mann–Whitney test. Source data are provided as a Source Data file.

## Methods

**Mice**. The _Foxp3YFP-cre_ mice (Jax No. 016959) are gifts from Prof. Xiaoming Feng (Peking Union Medical College, China).

The _Rbx1fl/fl_ mice were generated on the C57BL/6 background via CRISPR/ Cas9 technology with exons 2–4 floxed, performed by GemPharmatech Co., Ltd. (Nanjing, China). Briefly, the sgRNA (5′-A CCACTAAGTGGATAATCAC-3′ and 5′-GGAGTCAGAAATGATTGGTC-3′) direct Cas9 endonuclease cleavage in intron1–2 and intron 4–5 to insert LoxP sites by homologous recombination. The PCR genotyping primers 5′-GATTCTACTTTGCTTGCAGTGCTC-3′ and 5′-TCTGCATAAGCACGGGCTCTC-3′ generated DNA fragment of 266 and 196 bp bands, respectively, for mutant and wild type.

The _Ube2mfl/fl_ mice were generated on the C57BL/6 background, performed by GemPharmatech Co., Ltd. (Nanjing, China). The CRISPR/Cas9 technology was used to flox exons 2–4 with exon 5 frame-shifted in _Ube2m_ genome. Given exon 1 only encodes 36 amino acids, there is no possibility that a truncated functional protein is still being made. Briefly, the sgRNA (5′-TCAATTTAAGCTACCATAC-3′ and 5′-AGTGAAGATACGGGACA-3′) direct Cas9 endonuclease cleavage in intron1–2 and intron 4–5 to insert LoxP sites by homologous recombination. The

PCR genotyping primers 5′-CAAGACCTGCCTTCCAGGTATC-3′ and 5′-CCCTGCTAATACTGAACAGG-3′ generated DNA fragment of 317 and 223 bp for mutant and wild-type, respectively.

The _Ube2ffl/fl_ mice were generated from an ES clone (HEPD0820_4_G09), purchased from EuMMCR (European Mouse Mutant Cell Repository) (MGI:1915171) with exon 5 floxed. For LoxP sites flanked allele, the PCR genotyping primers 5′-GCGAGCTCAGACCATAACTTCG-3′ and 5′-CCAGGGTGGAAAATTTCAGTTT-3′ generated DNA fragment of 320 bp; and for wild type allele, the genotyping primers 5′-CCCTGGAATTTCGGTATTATA-3′ and 5′-CCAGGGTGGAAAATTTCAGTTT-3′ generated DNA fragment of 320 bp, respectively.

The _Sagfl/fl_ mice were generated and characterized as described[72].

The _Bimfl/fl_ mice were purchased from GemPharmatech Co., Ltd. (Nanjing, China) (T007185).

The _Rag1−/−_ mice were purchased from GemPharmatech Co., Ltd. (Nanjing, China) (T004753).

All mice were maintained in specific pathogen-free (SPF) conditions with experimental/control groups co-housed, and experiments were conducted

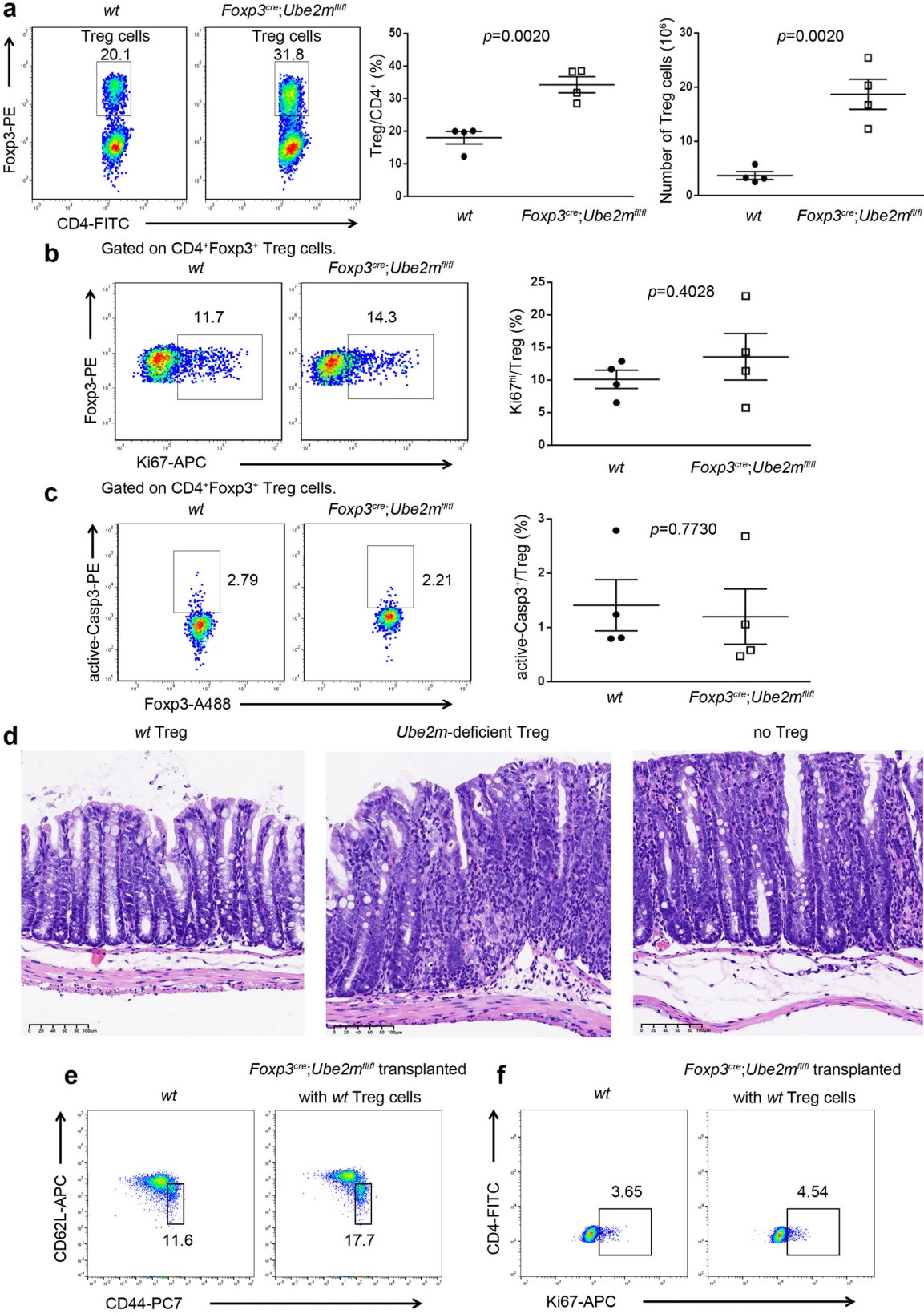

with gender-matched experimental/control mice. Mice were euthanized via cervical dislocation or carbon dioxide. Permission was granted to perform animal experiments by the Animal Ethics Committee of Zhejiang University; animal care was provided in accordance with the principles and procedures by the regulatory standards at Zhejiang University Laboratory Animal Center. To request the mouse strains generated in this project, please contact Dr. Yi Sun, e-mail: yisun@zju.edu.cn.

**Flow cytometry**. Peripheral lymph nodes and/or spleens were harvested from mice with indicated genotypes, and grinded into single cells, subjected to followed experiments. For analysis of surface markers, cells were washed in PBS containing 2% (wt/vol) FBS, and stained with indicated Abs (dilution 1: 200); intracellular markers were strained with Transcription Factor Buffer Set (BD Pharmingen, 562574) according to the manufacturer's instructions by indicated Abs (dilution 1: 50). For intracellular cytokine staining, cells were stimulated for 5 h with leukocyte

**Fig. 8 Impaired suppressive function of Ube2m-deficient Treg cells. a** Treg/CD4[+] ratios and Treg cell numbers in peripheral lymph nodes from *wt* and *Foxp3[cre];Ube2m[fl/fl]* mice (16 weeks old, *n* = 4 biologically independent samples from both male and female mice). **b** Ki67 in Treg cells in peripheral lymph nodes from *wt* and *Foxp3[cre];Ube2m[fl/fl]* mice (16 weeks old, *n* = 4 biologically independent samples from both male and female mice). **c** Active Caspase-3 in Treg cells in peripheral lymph nodes from *wt* and *Foxp3[cre];Ube2m[fl/fl]* mice (16 weeks old, *n* = 4 biologically independent samples from both male and female mice). **d** Representative images of distal colon after H&E staining (scale bar = 100 μm, male, three times each experiment was repeated independently with similar results). **e** Expression of CD44 and CD62L in Tcon cells from peripheral lymph nodes of *wt* and *Foxp3[cre];Ube2m[fl/fl]* (transplanted with *wt* Treg cells) mice (16 weeks, male). **f** Expression of Ki67 in Tcon cells from peripheral lymph nodes of *wt* and *Foxp3[cre];Ube2m[fl/fl]* (transplanted with *wt* Treg cells) mice (16 weeks, male). All error bars represent the SEM, data are presented as mean values +/− SEM. The *p* values were calculated by Mann–Whitney test. Source data are provided as a Source Data file.

---

activation cocktail of GolgiPlug (BD Pharmingen, 550583). Antibodies were all from eBioscience, unless otherwise noted: anti-CD4 (GK1.5), anti-CD8α (53-6.7, BD Pharmingen), anti-CD44 (IM7, BD Pharmingen), anti-CD62L (MEL-14), anti-Foxp3 (FJK-16s), anti-Ki67 (SolA15), anti-CD45 (30-F11), anti-F4/80 (BM8), anti-CD11b (M1/70), anti-CD11c (N418), anti-MHCII (M5/114.15.2), anti-CD80 (16-10A1), anti-CD86 (GL1, BD Pharmingen), anti-B220 (RA3-6B2), anti-CD19 (1D3, BD Pharmingen), anti-GL7 (GL7, BD Pharmingen), anti-Fas (15A7), anti-active caspase 3 (C92-605, BD Pharmingen), anti-Ifn-γ (XMG1.2), anti-Il4 (11B11), anti-Il17A (eBio17B7), and anti-Bim (C34C5, Cell Signal Technology). Flow cytometry data were acquired on CytoFLEX LX (Beckman) and analyzed using Flowjo software (Tree Star).

**Cytokines assay.** The serum was drawn from mice of indicated genotypes, diluted four-fold by PBS, and then subjected to cytokine/chemokine profiling using Mouse Cytokine Grp I (Bio-Plex Pro, M60009RDPD), measured on Luminex 200 system.

**Antibodies assay.** The immunoglobulin subclasses in mouse serum were measured using Mouse Immunoglobulin Isotyping Kit (BD Cytometric Bead Array, 550026).

**Cell sorting.** Peripheral lymph nodes and/or spleens were harvested from mice with indicated genotypes, and grinded into single cells, followed by treatment with Dynabeads™ Untouched™ Mouse CD4 Cells Kit (Invitrogen, 11415D) to isolate the CD4[+] T cells. The FACS sorting (SONY Cell Sorter, SH800S) was used to isolate the CD4[+]YFP[+] Treg cells with purities >99%.

**The in vivo immune-suppression assay in *Rag1[−/−]* mice.** Naive T cells sorted by FACS (CD4[+]YFP[−]CD44[lo]CD62L[hi], 4 × 10⁵) from *Foxp3[cre]* mice were transferred into *Rag1[−/−]* mice alone or in combination with indicated Treg cells (CD4[+]YFP[+], 2 × 10⁵). Mice were euthanized at 4 weeks after T-cell transfer and the colon tissues were collected for histopathological analysis. The severity of colon damages were classified into five categories as described previously[58]: Grade 0 has normal colonic crypt architecture; Grade 1 has mild inflammation with slight epithelial cell hyperplasia and increased numbers of leukocytes in the mucosa; Grade 2 has moderate colitis with pronounced epithelial hyperplasia, significant leukocyte infiltration, and decreased numbers of goblet cells; and Grade 3 has severe colitis with marked epithelial hyperplasia, extensive leukocyte infiltration, significant depletion of goblet cells, and occasional ulceration, or cryptic abscesses; Grade 4 has very severe colitis with marked epithelial hyperplasia, extensive transmural leukocyte infiltration, severe depletion of goblet cells, severe ulceration, and many crypt abscesses.

**The in vivo adaptation of *wt* Treg cells.** The CD4[+]YFP[+] *wt* Treg cells were sorted from the *Foxp3[cre]* mice, and transplanted to the recipient by intraperitoneal injection. For *Foxp3[cre];Rbx1[fl/fl]* mice, 4 × 10⁵ *wt* Treg cells were transplanted to the *Foxp3[cre];Rbx1[fl/fl]* mice at p8 stage, and the recipients were euthanized at p20 for analysis. For *Foxp3[cre];Ube2m[fl/fl]*, 8 × 10⁵ *wt* Treg cells were transplanted to the *Foxp3[cre];Ube2m[fl/fl]* mice at 6 weeks old, and the recipients were euthanized at 16-week old for analysis.

**Single-cell RNA sequencing.** CD4[+]YFP[+] Treg cells were isolated from the peripheral lymph nodes of mice with indicated genotypes. To generate enough materials, the lymph nodes from 3 *Foxp3[cre/wt]* or 6 *Foxp3[cre/wt];Rbx1[fl/fl]* mice at age of 10 weeks were pooled and subjected to single-cell RNA sequencing by Sinotech Genimics Co., Ltd. (Shanghai, China). The single-cell capture was achieved by random distribution of a single-cell suspension across >200,000 microwells through a limited dilution approach, and the *wt* and *Rbx1*-deficient Treg cells were distributed on two individual microwell plates. Beads with oligo-nucleotide barcodes were added to saturation so that a bead was paired with a cell in a microwell. Cell-lysis buffer was added to facilitate poly-adenylated RNA molecules hybridized to the beads. Beads were collected into a single tube for reverse transcription. Upon cDNA synthesis, each cDNA molecule was tagged on the 5′ end (that is, the 3′ end of a mRNA transcript) with a molecular index and cell label to indicate its cell of origin. Whole transcriptome libraries were prepared using the BD Resolve system for single-cell whole-transcriptome amplification workflow (BD Genomics) to capture transcriptomic information of the sorted single cells. Briefly, the second strand cDNA was synthesized, followed by ligation of the adaptor for universal amplification. Eighteen cycles of PCR were used to amplify the adaptor-ligated cDNA products. Sequencing libraries were prepared using random priming PCR of the whole-transcriptome amplification products to enrich the 3′ end of the transcripts linked with the cell labels and molecular indices. Sequencing libraries were quantified using a High Sensitivity DNA chip (Agilent) on a Bioanalyzer 2100 and the Qubit High Sensitivity DNA assay (Thermo Fisher Scientific). Approximate 1.5 pM of the library for each sample was loaded onto a NextSeq 500 system and sequenced using High Output sequencing kits (75 × 2 bp) (Illumina). After discarding the cells with high mitochondrial gene, 10,048 and 9736 cells were identified for *wt* and *Rbx1*-deficient Treg cells, with an average number of reads of 23,321.27 vs. 22202.55, and 1080.16 vs. 870.28 of genes detected per cell, respectively.

**Transcriptome profiling.** CD4[+]YFP[+] Treg cells were isolated from the peripheral lymph nodes and spleens of mice (8–12 weeks old) with indicated genotypes. To generate enough materials, the pooled tissues from 2 to 3 mice (*Foxp3[cre/wt]*), 8 to 11 (*Foxp3[cre/wt];Rbx1[fl/fl]*) or 3 to 4 (*Foxp3[cre/wt];Ube2m[fl/fl]*) were used. RNA samples were prepared with the miRNeasy Mini Kit (Qiagen, 21704), then reverse transcribed, amplified, labeled (Affymetrix GeneChip pico kit, 703308), and hybridized to Clariom S Arrays, mouse (Affymetrix, 902931), performed by Cnkingbio Biotech Co., Ltd. (Beijing, China). Microarray data sets were analyzed with Applied Biosystems Expression Console Software 1.4. Three independent experiments were performed.

**Proteomics profiling.** CD4[+]YFP[+] Treg cells were isolated from the peripheral lymph nodes and spleens of mice (>8 weeks of age) with indicated genotypes. To secure enough samples, tissues from 5 to 6 of *Foxp3[cre/wt]* or 21 to 24 of *Foxp3[cre/wt];Rbx1[fl/fl]* mice were pooled, washed with PBS buffer, and stored at −80 °C. Three independent collected pooled samples were subjected to Mass-Spectrometry.

Cells were resuspended in 5 μl 1% (w/v) sodium deoxycholate, 10 mM TCEP, 40 mM 2-chloroacetamide (CAA), 100 mM Tris (pH 8.5), and subsequently lysed by 5 min boiling at 95 °C and sonication (Bioruptor, model UCD-200, Diagenode) for 15 min. Cell debris were pelleted by centrifugation at 13,200r.p.m. for 5 min and the clarified lysate was transferred into a new vial. The lysate was diluted 1:10 for LysC-trypsin digestion (0.4 μg of each enzyme in double distilled water), and the digestion was performed overnight at 37 °C. The digest was acidified with 50 μl 2% TFA and sodium deoxycholate was extracted using 50 μl ethyl acetate and vigorous shaking. The organic phase was removed after centrifugation at 13,200r.p.m. for 5 min. Finally, the peptides were desalted on C18 StageTips.

Samples were analyzed on Orbitrap Fusion Lumos mass spectrometers (Thermo Fisher Scientific, Rockford, IL, USA) coupled with an Easy-nLC 1000 nanoflow LC system (Thermo Fisher Scientific). Dried peptide samples were re-dissolved in Solvent A (0.1% formic acid in water) and loaded to a trap column (100 μm × 2 cm, home-made; particle size, 3 μm; pore size, 120 Å; Dr. Maisch, Ammerbuch, Germany) with a max pressure of 280 bar using Solvent A, then separated on a home-made 150 μm × 30 cm silica microcolumn (particle size, 1.9 μm; pore size, 120 Å; Dr. Maisch, Ammerbuch, Germany) with a gradient of 5–35% mobile phase B (acetonitrile and 0.1% formic acid) at a flow rate of 600 nl/min for 141 min then up to 95% in 1 min and eluted for 9 min (the column temperature was maintained at 60 °C). The MS analysis were performed in data-dependent acquisition mode. Automatic gain control (AGC) targets were 5 × 10⁵ ions with a max injection time of 50 ms for full scans and 5 × 10³ with 35 ms for MS/MS scans. The most intense ions selected under top-speed mode were isolated in Quadrupole with a 1.6 m/z window and fragmented by higher energy collisional dissociation (HCD) with normalized collision energy of 32% and dynamic exclusion time was set as 25 s. Data were acquired using the Xcalibur software (Thermo Fisher Scientific).

DAVID was used for pathway enrichment analysis of proteins with change fold >2/3 or <0.5/0.33. The enriched pathways and their contained proteins with two-fold change were shown in heat map in which each quantitative value was normalized based its maximum value.

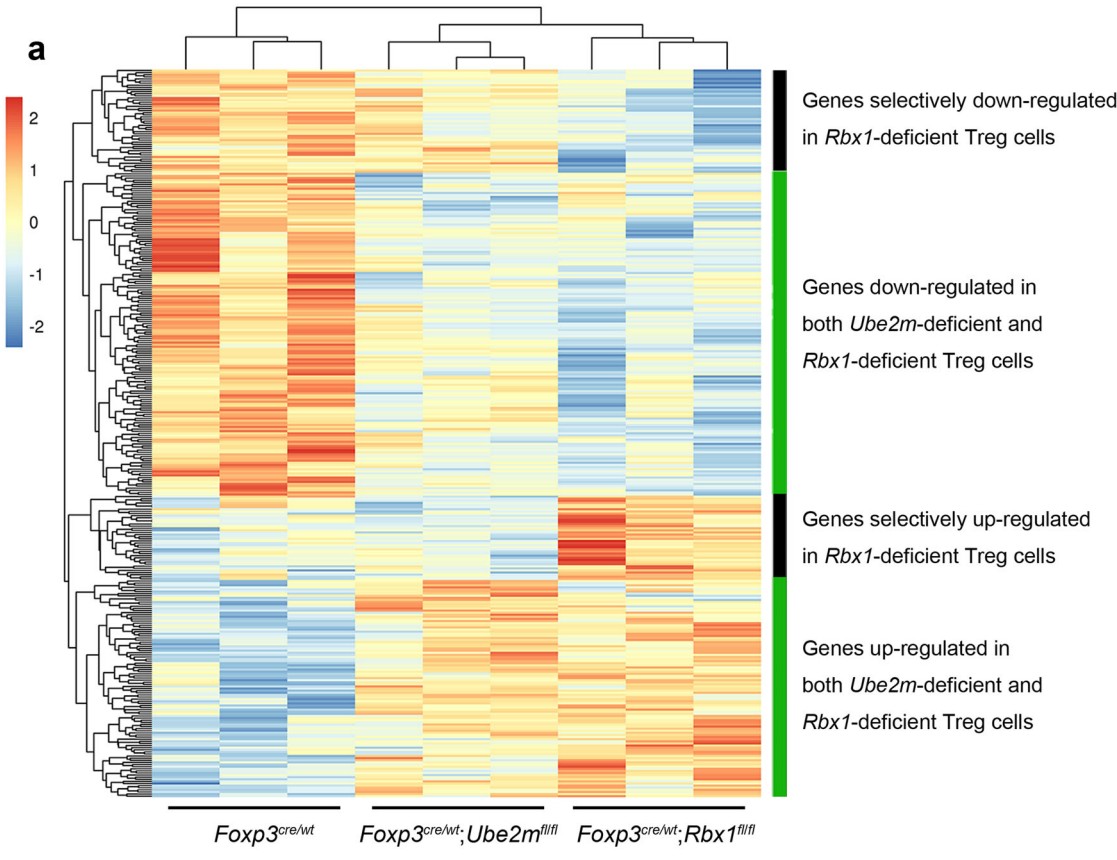

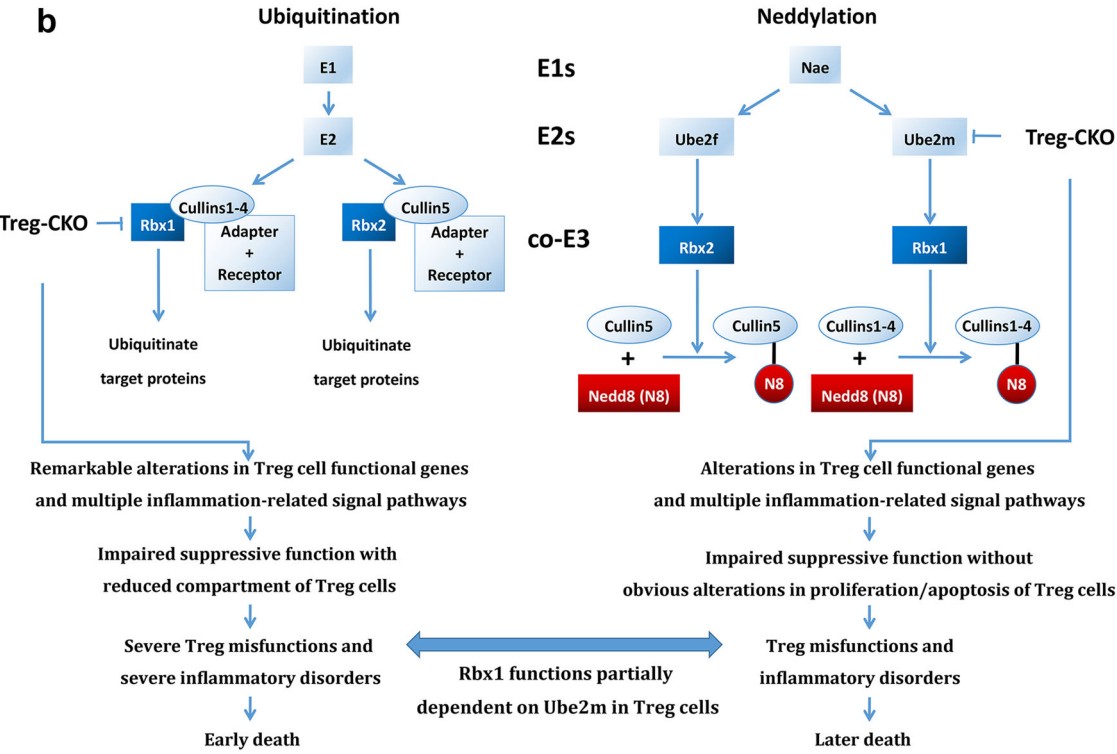

**Fig. 9 Rbx1 acts via Ube2m-dependent and -independent mechanisms in Treg cells. a** Unbiased cluster analysis of the transcriptional programs revealed four categories of genes differentially expression in *Rbx1*- and *Ube2m*-deficient Treg cells, as compared to the *wt* control Treg. **b** Working model of Rbx1 and Ube2m in functional regulation of Treg cells. The Treg specific *Rbx1* deletion causes the alterations in Treg functional genes, and multiple inflammatory and signaling, leading to early-onset immune disorder and death, whereas *Ube2m* deletion in Treg cells also caused altered inflammatory signaling pathway with much less extent, and death at the later stage. Lack of phenotype by deletion of *Rbx2/Sag* or *Ube2f* in Treg cells excludes the possible involvement of CRL5 in Treg regulation.

**RNA and immunoblot analysis**. The sequences of the primer pairs used in qPCR are as follows: *Sag*-Fwd: 5′-TG GAGGACGGCGAGGAAC-3′, *Sag*-Rev: 5′-CCCCAGA CCACAACACAGTC-3′; *Rbx1*-Fwd: 5′-CTTTGTATCGAATGTCAGGC-3′, *Rbx1*-Rev: 5′-GTCACTAGACGAGTAACAG-3′; *Ube2f*-Fwd: 5′-GACTGTAAGCCC AGATGAG-3′, *Ube2f*-Rev: 5′-CCTTTAATGTTCTAGTGGG-3′; *Ube2m*-Fwd: 5′-CTGTCCTGATGAAGGCTTC-3′, *Ube2m*-Rev: 5′-GTTCGGCTCCAAGAAGAG-3′; *Gapdh*-Fwd: 5′-GCCGCCTGGAGAAACCTGCC-3′, *Gapdh*-Rev: 5′-GGTGGAAG AGTGGGAGTTGC-3′.

The antibodies used in western blot are as follows: anti-Ube2f, Proteintech, 17056-1-AP (dilution 1: 1000); and anti-β-Actin, HuaBio, M1210-2 (dilution 1: 5000).

**Statistical analysis**. The $p$ values were calculated by Mann–Whitney test, two-sided unpaired Student's t-test, one-way ANOVA or two-way ANOVA as indicated using GraphPad Prism, unless otherwise noted. Statistical analysis of mouse survival and respective $p$ values were determined using the log-rank test. $p < 0.05$ was considered as significant. All error bars represent the SEM from three independent experiments.

**Reporting summary**. Further information on research design is available in the Nature Research Reporting Summary linked to this article.

## Data availability

The scRNA-seq and microarray data generated in this study have been deposited in the Gene Expression Omnibus under accession code GSE199289 and GSE192664. The mass spectrometry proteomics data generated in this study have been deposited in the ProteomeXchange Consortium via the iProX partner repository[73] under accession code PXD030144. Source data are provided with this paper.

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

# ARTICLE

47. Zhang, C. et al. Cullin3-KLHL25 Ubiquitin ligase targets ACLY for degradation to inhibit lipid synthesis and tumor progression. *Genes Dev.* **30**, 1956–1970 (2016).

48. Ambrosini, G., Seelman, S. L. & Schwartz, G. K. Differentiation-related gene-1 decreases Bim stability by proteasome-mediated degradation. *Cancer Res.* **15**, 6115–6121 (2009).

49. Lu, G. et al. Phosphorylation of ETS1 by Src family kinases prevents its recognition by the COP1 tumor suppressor. *Cancer Cell* **26**, 222–234 (2014).

50. Nakayama, K. et al. Impaired degradation of inhibitory subunit of NF-kappa B (I Kappa B) and Beta-Catenin as a result of targeted disruption of the beta-TrCP1 gene. *Proc. Natl Acad. Sci. USA* **100**, 8752–8757 (2003).

51. Zhang, J. et al. VHL substrate transcription factor ZHX2 as an oncogenic driver in clear cell renal cell carcinoma. *Science* **6399**, 290–295 (2018).

52. Huang, H. et al. Skp2 inhibits FOXO1 in tumor suppression through ubiquitin-mediated degradation. *Proc. Natl Acad. Sci. USA* **102**, 1649–1654 (2005).

53. Yang, K. et al. Homeostatic control of metabolic and functional fitness of Treg cells by LKB1 signalling. *Nature* **548**, 602–606 (2017).

54. O'Connor, L. et al. Bim: A novel member of the Bcl-2 family that promotes apoptosis. *EMBO J.* **17**, 384–395 (1998).

55. Chougnet, C. A. et al. A major role for Bim in regulatory T cell homeostasis. *J. Immunol.* **186**, 156–163 (2011).

56. Zhao, Y., Morgan, M. A. & Sun, Y. Targeting Neddylation pathways to inactivate cullin-RING ligases for anticancer therapy. *Antioxid. Redox Signal.* **21**, 2383–2400 (2014).

57. Deshaies, R. J. SCF and Cullin/Ring H2-based ubiquitin ligases. *Annu. Rev. Cell Dev. Biol.* **15**, 435–467 (1999).

58. Ouyang, W. et al. Novel Foxo1-dependent transcriptional programs control $T_{reg}$ cell function. *Nature* **491**, 554–559 (2012).

59. Zeng, H. et al. mTORC1 couples immune signals and metabolic programming to establish $T_{reg}$-cell function. *Nature* **499**, 485–490 (2013).

60. Han, X. et al. Mapping the mouse cell atlas by microwell-seq. *Cell* **172**, 1091–1107 (2018).

61. Weinberg, S. E. et al. Mitochondrial complex III is essential for suppressive function of regulatory T cells. *Nature* **565**, 495–499 (2019).

62. Su, W. et al. Protein Prenylation Drives Discrete Signaling Programs for the Differentiation and Maintenance of Effector Treg Cells. *Cell Metab.* **32**, 996–1011 (2020).

63. Ivan, M. et al. HIFalpha targeted for VHL-mediated destruction by proline hydroxylation: Implications for $O_2$ sensing. *Science* **292**, 464–468 (2001).

64. Wang, D. et al. A deep proteome and transcriptome abundance atlas of 29 healthy human tissues. *Mol. Syst. Biol.* **15**, e8503 (2019).

65. Jiang, L. et al. A quantitative proteome map of the human body. *Cell* **20**, 31078–31083 (2020).

66. Cuadrado, E. et al. Proteomic analyses of human regulatory T cells reveal adaptations in signaling pathways that protect cellular identity. *Immunity* **48**, 1046–1059 (2018).

67. Chang, M., Lee, A. J., Fitzpatrick, L., Zhang, M. & Sun, S. C. NF-kappa B1 p105 regulates T cell homeostasis and prevents chronic inflammation. *J. Immunol.* **182**, 3131–3138 (2009).

68. Tong, J. et al. m6A mRNA methylation sustains Treg suppressive functions. *Cell Res.* **28**, 253–256 (2018).

69. Zhang, M. et al. The roles of Egr-2 in autoimmune diseases. *Inflammation* **38**, 972–977 (2015).

70. Iellem, A. et al. Unique chemotactic response profile and specific expression of chemokine receptors CCR4 and CCR8 by $CD4^+CD25^+$ regulatory T cells. *J. Exp. Med.* **194**, 847–853 (2001).

71. Joller, N. et al. Treg cells expressing the coinhibitory molecule TIGIT selectively inhibit proinflammatory Th1 and Th17 cell responses. *Immunity* **40**, 569–581 (2014).

72. Li, H. et al. Inactivation of SAG/RBX2 E3 ubiquitin ligase suppresses KrasG12D-driven lung tumorigenesis. *J. Clin. Invest.* **124**, 835–846 (2014).

73. Ma, J. et al. iProX: An integrated proteome resource. *Nucleic Acids Res.* **47**, D1211–D1217 (2019).

## Acknowledgements

We would like to thank Prof. Xiaoming Feng for providing *Foxp3YFP-cre* mice. We also thank the Core Facilities, Zhejiang University School of Medicine for the technical support. This project was supported by the National Key R&D Program of China (2021YFA1101000 and 2016YFA0501800 to Y.S.), National Natural Science Foundation of China (82172699 and 81801567 to D.W.), and Zhejiang Provincial Natural Science Foundation of China (LY21H100005 to D.W. and LD22H300003 to Y.S.).

## Author contributions

D.W. designed and preformed experiments, analyzed data, and wrote the manuscript. H.L. analyzed proteome and transcriptome data. M.L. performed the mass spectrum experiment and analyzed the data. J.Q. designed and supervised the mass spectrum experiment and analyzed data. Y.S. designed experiments, analyzed data, wrote and finalize the manuscript, and oversaw the project.

## Competing interests

The authors declare no competing interests.
