## [Peer Review File · Nature Communications]

The Ube2m-Rbx1 neddylation-Cullin-RING-Ligase proteins are essential for the maintenance of Regulatory T cell fitnessReviewers' comments:

Reviewer #1 (Remarks to the Author):

There is increasing interest in post translational modifications in the regulation of immune cell physiology and function. Several papers have been published in recent years about the role of ubiquitination in regulatory T cells (Tregs), especially in controlling protein levels of FoxP3, the major transcription factor that governs much of Treg function. Also, the Cullin RING ligase VHL, which regulates degradation of the HIF1a protein, was shown to have a critical role in Tregs. The current manuscript by Wu et al. fits nicely in this theme. In particular, they show through genetic deletion that Rbx1, one of the two catalytic subunits used by the RING family of Ubiquitin ligases, is required for proper Treg function in mice. The other catalytic subunit, Rbx2, has an important function in conventional T cells, but seems dispensable for Tregs. Finally, they show that Ube2M, a protein involved in activation of Cullin RING ligases by Neddylation, is also necessary for Treg function, although genetic loss of the gene encoding this protein does not phenocopy loss of Rbx1, suggesting a layer of complexity in control of RING ligases that is not fully understood.

Although there has been some attention in the field for protein ubiquitination in Tregs, the findings reported in this manuscript are certainly novel and of interest to the Treg community. Since there are unexpected discrepancies in the phenotypes between deficiency for Ube2m and Rbx1, the study may also be of interest to investigators in the ubiquitin ligase and neddylation field. The results reported are convincing and the phenotypes reported are striking. Still, I found myself wondering what the real message of this paper is. At this stage, the manuscript is very descriptive and, other than that these two particular genes apparently have important roles in Tregs, I did not come away from reading the manuscript with the conviction that I now understood something that I did not understand before. The single cell RNAseq experiment nicely shows that Rbx1 is necessary specifically for the generation and/or maintenance of a cohort of so called effector Tregs, activated cells that may be the most suppressive types of Tregs and that go into tissues to exert their functions. How does Rbx1 do this? Is Rbx1 specifically expressed in effector Tregs? Is it specifically active in effector Tregs? Or is the regulation on its targets? The study suggests, but does not show, that Rbx1 may regulate stability of the pro-apoptotic Bim protein. If we assume that this is indeed the mechanism, what does this then teach us? Is this Bim protein only expressed in effector Tregs? How is Bim turned into a target for Rbx1 and what is the use of this regulatory mechanism?

Apart from this larger conceptual consideration, I wonder about the logic of the bulk proteome-transcriptome comparison in figures 4 and 5. The single cell RNAseq experiment very convincingly shows that Rbx1 deficiency results in loss of a population of cells. A bulk transcriptome analysis of the entire Treg populations in wild type and knock out mice therefore compares not only the direct consequence of Rbx1 deficiency, but also the indirect effect of loss of a whole cohort of cells. This makes the interpretation of the results very difficult, it seems to me.

<minor comments>

1. I found the statement in the introduction (line 47/48) that "very limited numbers of genes have been shown to play essential roles in Tregs" a little misleading. The list of genes with proven essential roles in Tregs is quite long and comprises many more genes than the ones listed here.
2. It would be good to mention the gender of the mice used. Since the FoxP3 gene is X-linked, this is important. In fact, this is essential for the interpretation of the RNAseq results, which were performed on (I surmised) female mice, which are heterozygous for the FoxP3-Cre knock in allele, such that 50% of their cells are wild type. Although Supplementary figure 5 does explain this, I think this can be made much clearer by mentioning the genders and perhaps adding a sentence in the main body of the manuscript about the X-chromosomal location of the FoxP3 gene and the fact that female mice are chimeric.
3. It was not clear to me which of the genes/proteins shown in the heat maps in figures 4 and 5 are significantly differentially expressed. This should be clearly indicated. How was the selection of genes shown here made?

4. It is surprising that the Treg-specific Rbx2 and Ube2f deficient mice do not have a Treg phenotype. I suppose the Rbx2 mice were confirmed previously to lack expression of this protein, (might be good to point that out), but I did not see evidence that the Ube2f targeting led to loss of expression. As for the Ube2m deficient mice: is there a possibility that a truncated protein is still made, or perhaps that a splice variant is expressed that lacks the targeted exons, but still has partial function? If that is the case, it will alter the interpretation of the different phenotypes of the Rbx1 and the Ube2m deficiencies.

In general, the characterization of the mice is a little thin. Only mRNA expression levels are shown, which are not necessarily a good reflection of functional gene deficiency. Do the RT-PCR primers used for analysis anneal in the exons between the LoxP sites? If so, alternative transcripts may still be made. Minimally, it would be helpful to mention the sequences of the primers and to mention where they anneal. In addition, it might be a good idea to test expression of the proteins (this seems especially relevant for the Ube2m and Ube2f proteins), ideally with antibodies to the N- and to the C-termini of the proteins.

Reviewer #2 (Remarks to the Author):

Wu et al

Requirement of Rbx1 Cullin-Ring ligase E3 and Ube2m neddylation E2 in the maintenance of the fitness of Treg cells

Here the authors investigate the effect of a Treg-specific deletion of the Tbx1 component of cullin-RING ligases. This results in early onset fetal inflammatory disorders, disrupted homeostasis and impaired Treg functions. Similar phenotypes were seen in mice with Ube2m, a neddylation E2 ligase in Treg cells but with less severity. They propose that the Ube2-Rbx1 axis is required for the maintenance and homeostasis of Treg cells. They further propose that pevonedistat, a potent inhibitor of neddylation/CRLs, may have therapeutic value.

General

The CRL family is large. Is there any specificity in expression of components in Treg cells versus other effector populations? It would be interesting, and therapeutically important, to know if Ube2-Rbx1 is Treg cell specific.

The authors should demonstrate Rbx1- and Ube2- deletion in Treg cells by an alternative approach than relative levels of mRNA by sequencing. Since this is the focus of the whole paper, it would be appropriate here to show both genomic deletion (qRT-PCR) and western blot analysis.

In figures where ratios and percentages of immune cells are shown, there should also be information concerning absolute numbers to get a better idea of the effects of Rbx1-deletion on immune homeostasis.

I struggle somewhat to understand the specificity and function of Rbx1 in Treg cells. There are dramatic transcriptional changes occurring after deletion of Rbx1, and also proteomic changes. This seems to affect pretty much every major pathway that could be envisaged. Is this not simply because the cells are dying (evident from greatly reduced numbers, reduced proliferation and increased apoptosis)? This really needs to be discussed in detail and some mechanistic insight presented. The study is in general very descriptive and speculative, leaning on a lot of transcriptome analyses, and mechanistically weak. Is the response of Treg cells to loss of Rbx1 not just a response to induction of cell death in the context of an immune system? How is Rbx1-deletion so dramatically affecting the transcriptome and proteome?

Specific

Figure 1. It would be appropriate to not only provide information concerning CD4+ and CD8+ cells, but also B cells (CD19), DCs (CD11c), macrophages (CD11b) etc to enable a proper understanding of the effect of Rbx1 deletion on immune homeostasis.

Figure 2. The authors conclude these experiments by stating that Rbx1 is needed for the suppressive function of Treg cells. However, they show that Rbx1-deletion leads to greatly reduced numbers of Treg cells due to reduced proliferation (Fig 2b) and increased apoptosis (Fig 2c). I find it difficult to conclude that it is Treg-suppressive function that has been affected. To confirm this they should perform an in vitro or in vivo suppression assay.

Figure 3. The authors should run some direct transcriptome comparisons between clusters 0, 3 and 4 and online (effector) Treg datasets rather than handpicking some Treg-related genes. How similar are these clusters to predefined PB or tissue-resident Treg cell transcriptomes? To conclude that "Rbx1 is essential for maintenance of effector subpopulations of Treg cells" it would be relevant to confirm this using FACS-based technologies and not just sc-RNA-seq.

Figure 4. In Figure 4a-b the authors list a set of important genes for Treg cell function whose expression is changed by loss of Rbx1. It seems that very few of these actually demonstrate changes on the protein level (as evidenced by the mass spec data). What is the relevance of the data presented in Figure 4b if the protein level of these genes is not affected by Rbx1-deletion? If the authors wish to propose that Rbx1 is involved in energy and nucleotide metabolism then they should provide some experimental data to support this.

Figure 6. As also mentioned above, the authors should do a proper characterization of immune cell types and numbers here, and also compare this to Rbx1-deletion. Just ratios of Treg/CD4 are not sufficient to understand the phenotype. Can the authors explain the increased number of Treg cells in the LN? Are the Ube2m-deleted Treg cells able to suppress effector proliferation in vitro or in vivo?

Minor

The introductory paragraph does not really give justice to the field. It suggests there are a small number of genes with essential roles in Treg biology and that "studies are rather limited as to how Treg cells are precisely regulated". There is in fact a body of literature reporting studies investigating Treg cell biology. I think the authors could better just focus on a lack of information concerning CRLs in Treg cell biology rather than this general statement.

Figure 5b. Presenting one FACS blot is not sufficient, there should also be quantification of multiple experiments.

The authors should remove the discussion of pevonedistat from the abstract and introduction as they do not perform any experiments showing that this affects Treg cell function. They state in the discussion that "our study provides a proof-of-concept evidence that pevonedistat may have novel application in the treatment of diseases associated with over-activation of Treg cells". I don't agree with them that they have really provided proof-of-concept and this should be rephrased. It would certainly be interesting, and also improve the relevance of the manuscript, if they could perform in vivo, or minimally in vitro experiments to validate this hypothesis.

Reviewer #3 (Remarks to the Author):

The authors build on their earlier studies Rbx1 and Sag2/Rbx2 whole animal knockouts. Here they delete Rbx1, Sag2/Rbx2, Ube2m, or Ube2f in Tregs using the Foxp3cre driver. These genes encode critical components of CRL E3 multi-unit ubiquitin ligases and neddylation regulatory pathway proteins. UBE2M with RBX1 serve in neddylation of CUL1-CUL4. UBE2F with SAG2/RBX2 serve in neddylation of CUL5. The authors perform several standard assays in the field to examine effects

on animal physiology and proliferation, viability, and function of Tregs in the knockouts. The deletion of SAG2/RBX2 and UBE2F have no obvious effects. The deletion of RBX1 is reminiscent of loss of Foxp3 and generally reflects RBX1 as essential for Tregs in vivo. The deletion of Ube2m has an intriguing phenotype. The mice suffer from inflammatory disorders, but not with the same severity, rate, or penetrance of Rbx1. From an immunological perspective, the results do not seem surprising and scale with the known essentiality of these genes for proliferation.

For molecular analyses, the authors perform scRNA-seq and proteomics on the Rbx1 null Tregs, with some western blots, and transcriptome analyses on the Ube2m nulls. They conclude that Rbx1 plays Ube2m-dependent and -independent roles in Treg cells.

Major comments:

1. In terms of the experiments, the authors should do the control experiments for Rbx1 and Ube2m and transfer Tregs from WT mice into the newborn mutants to be sure they rescue the immune phenotypes.
2. At this point, there have been so many high throughput CRISPR screens defining essential gene sets and the effects on Tregs scale well with the predicted essentiality of the genes. Rbx1 has been defined as amongst the most essential genes for cell proliferation and the phenotype may well reflect this instead of an effect on specific substrates. In this regard, it is not clear that the proteomics adds the value implied for identifying substrates of Rbx1 E3 ligase with a Treg specific function but more reflects essentiality of Rbx1. Without substantial additional effort at validation, it is not clear what the proteome data provide in terms of mechanism. The conclusion that the proteomic differences are true substrates should be toned down.
3. The interesting results come from the comparison to Ube2m. It is unfortunate for the authors that some people might expect Ube2m would have milder effects because Ube2f technically could compensate. But this reviewer thinks enough investigators would not have this depth of knowledge and would find this intriguing, if the mechanisms were clarified. Some mechanisms that come to mind would be if Ube2f is compensating. The authors must have made the double null already and this could be tested. Or if neddylation is provided by unknown enzymes, this could be tested by pharmacologic or genetic ablation of Nedd8 E1 activity. Another possibility is that neddylation is not so essential in Tregs, a result that could also come from the latter experiments.
4. The differences between the effects of deleting Rbx1 and Ube2m require more explanation. The Treg/CD4+ ratio for Rbx1 make sense but for Ube2m the ratio is increased compared to WT controls. This is counterintuitive and the authors should provide a mechanism.

Minor points:

1. This is not entirely the common view in the field and the authors should either remove this sentence or provide a more balanced perspective:
RBX1 binds to CDC34/UBCH2C to promote ubiquitylation of large number of substrates via the K48 linkage, whereas RBX2 binds to UBE2C/UBE2S to promote ubiquitylation of different set of substrates via the K11 linkage for proteasome degradation^{17,18}
2. Transcriptome is mis-spelled in figures.
3. The authors may wish to elaborate further on the implications of the Rbx2 and Ube2f nulls for Treg regulation by SOCS proteins.

Responses to the critiques from three insightful reviewers

Reviewer #1 (Remarks to the Author):

There is increasing interest in post translational modifications in the regulation of immune cell physiology and function. Several papers have been published in recent years about the role of ubiquitination in regulatory T cells (Tregs), especially in controlling protein levels of FoxP3, the major transcription factor that governs much of Treg function. Also, the Cullin RING ligase VHL, which regulates degradation of the HIF1a protein, was shown to have a critical role in Tregs. The current manuscript by Wu et al. fits nicely in this theme. In particular, they show through genetic deletion that Rbx1, one of the two catalytic subunits used by the RING family of Ubiquitin ligases, is required for proper Treg function in mice. The other catalytic subunit, Rbx2, has an important function in conventional T cells, but seems dispensable for Tregs. Finally, they show that Ube2M, a protein involved in activation of Cullin RING ligases by Neddylation, is also necessary for Treg function, although genetic loss of the gene encoding this protein does not phenocopy loss of Rbx1, suggesting a layer of complexity in control of RING ligases that is not fully understood.

Although there has been some attention in the field for protein ubiquitination in Tregs, the findings reported in this manuscript are certainly novel and of interest to the Treg community. Since there are unexpected discrepancies in the phenotypes between deficiency for Ube2m and Rbx1, the study may also be of interest to investigators in the ubiquitin ligase and neddylation field. The results reported are convincing and the phenotypes reported are striking.

We thank the reviewer for his/her comprehensive and excellent summary of our work and its importance to both fields of CRL-neddylation and Treg field.

Still, I found myself wondering what the real message of this paper is. At this stage, the manuscript is very descriptive and, other than that these two particular genes apparently have important roles in Tregs, I did not come away from reading the manuscript with the conviction that I now understood something that I did not understand before.

*We valued the comments from this thoughtful and critical reviewer. We should have emphasized more in our original submission that Rbx1 is indeed an essential gene for Treg cells, given the fact that mice with Rbx1-null Treg cells died within 30 days after the birth due to severe autoimmune disorders. This severe phenotype of robust inflammation observed in *Foxp3^{cre};Rbx1^{fl/fl}* mice is reminiscent of Treg-deficient mice (PMID: 17136045) or mice with loss-of-function mutations in the *Foxp3* gene (PMID: 17220874), suggesting *Rbx1* is absolutely essential for Treg cells *in vivo*. Such severe phenotype was only observed in Treg deletion of a handful of genes, including *Foxp3* (PMID: 12522256/12612578/12612581), *Foxo1* (PMID: 23135404), *Raptor* (PMID: 23812589), *Lkb1* (PMID: 28847007), *Uqcrsf1* (PMID: 30626970), and *Pggt1b* (PMID: 33207246). Thus,*

the finding that Rbx1 is such a critical gene for Treg cells should be considered as a significant progress in the field of Treg cell research. Furthermore, given Rbx1 is a dual E3 for both NEDD8 neddylation ligase and Cullin RING ubiquitin ligase (CRLs), our study also revealed the critical roles of both neddylation modification and CRLs for the functionality of Treg cells, which is previously unknown. We have added these comments in the Discussion section (Page 18).

Technically, we used in this study three comprehensive approaches, namely single cell sequencing, whole transcriptome profiling, and Mass-spect based proteomics in an attempt to define the mechanism of Rbx1 action in Treg cells. Given the broad role of the neddylation-CRLs in fundamental biological processes, our study yielded a wealth of information on many altered signaling pathways and individual gene/protein upon Rbx1 Treg deletion. We reason that this is the first overall phenotypical and global mechanistic study of elucidating the role of neddylation enzymes, particular Rbx1/Ube2m, in Treg cells, which sets the foundation for future more detailed mechanistic studies to pinpoint special role of each interesting pathways and individual genes in regulation of Treg functions. We have now added this paragraph in the Discussion section (Page 23).

The single cell RNAseq experiment nicely shows that Rbx1 is necessary specifically for the generation and/or maintenance of a cohort of so called effector Tregs, activated cells that may be the most suppressive types of Tregs and that go into tissues to exert their functions. How does Rbx1 do this? Is Rbx1 specifically expressed in effector Tregs? Is it specifically active in effector Tregs? Or is the regulation on its targets?

As shown below, single cell sequencing revealed that *Rbx1* gene is not specifically expressed in the effector Treg cells, rather it is expressed in all sub-populations of Treg cells (Supplementary Fig. 25a).

Decrease in the effector subpopulation (Clusters 3&4) and increase in the naive population (Cluster 0) of Treg cells upon *Rbx1* Treg deletion clearly indicated that Rbx1 depletion causes the failure in the conversion of subgroup 0 to subgroup 3/4, which is a very interesting and novel finding. As to how Rbx1 does it, we can only speculate at this stage. Given major role of Rbx1 as a dual E3 for both neddylation and ubiquitylation, it is speculated that Rbx1 depletion would cause abnormal levels of transcription factors

and/or signaling molecules directly (via substrate accumulation or neddylation modification) or indirectly (altered transcriptions due to direct effect). Future studies will be directed to elucidate the underlying mechanism. We have now discussed this important issue in the Discussion section (Page 20).

The study suggests, but does not show, that Rbx1 may regulate stability of the pro-apoptotic Bim protein. If we assume that this is indeed the mechanism, what does this then teach us? Is this Bim protein only expressed in effector Tregs? How is Bim turned into a target for Rbx1 and what is the use of this regulatory mechanism?

Pro-apoptotic protein Bim is a known substrate of Rbx1 (PMID: 23912711). Accumulation of Bim in Rbx1-null Treg cells suggest that Bim is likely responsible for enhanced apoptosis seen in Treg cells.

The single cell RNA sequencing data revealed that *Bim* mRNA is widely expressed in all the subpopulations of Treg cells (see below, also Fig. 5c), indicating that *Bim* expression is not specific to effector Tregs. The detection of increased Bim levels in Rbx1-null Treg cells with reduced Treg cell portions highly suggested that Bim is a Rbx1 substrate in Treg cells. We have now added these descriptions in the text (Page 13).

Apart from this larger conceptual consideration, I wonder about the logic of the bulk proteome-transcriptome comparison in figures 4 and 5. The single cell RNAseq experiment very convincingly shows that Rbx1 deficiency results in loss of a population of cells. A bulk transcriptome analysis of the entire Treg populations in wild type and knock out mice therefore compares not only the direct consequence of Rbx1 deficiency, but also the indirect effect of loss of a whole cohort of cells. This makes the interpretation of the results very difficult, it seems to me.

The reviewer made a valid point. The purpose for the bulk proteome-transcriptome comparison is to find out potential substrates of CRLs, which should be increased at the protein levels (Proteome), but not at the mRNA levels (transcriptome). We acknowledge that a bulk transcriptome analysis would detect both direct and indirect consequences of Rbx1 deficiency in Treg cells. The advantage of such an analysis is to detect unbiasedly all alterations at the whole transcriptome levels in Treg cells in response to *Rbx1* deletion.

<minor comments>

1. I found the statement in the introduction (line 47/48) that "very limited numbers of genes have been shown to play essential roles in Tregs" a little misleading. The list of genes with proven essential roles in Tregs is quite long and comprises many more genes than the ones listed here.

The reviewer made an excellent point, and we should have made our statement clearer. Since 2003 when *Foxp3* was identified as a specific marker of Treg cells, many investigators in the field had used the *Foxp3*^{Cre}-loxP system to selectively knock out the genes of their interest in mouse Treg cells in an attempt to find other key regulators. In general, if Treg deletion of a given gene reproduces the phenotype of Treg cell depletion, namely the mice develop a severe autoimmune response and die around 30 days, the gene in question is critically essential for Treg cells. Only a handful of genes meet this stringent criteria, including *Foxp3* (PMID: 12522256/12612578/12612581), *Foxo1* (PMID: 23135404), *Raptor* (PMID: 23812589), *Lkb1* (PMID: 28847007), *Uqcrsf1* (PMID: 30626970), and *Pggt1b* (PMID: 33207246). Our study identified *Rbx1* as an addition to this limited list.

The reviewer was correct that many other genes are also important and Treg deletion of these genes indeed caused inflammatory response, but to lesser extent with mice survived 70 days or beyond. We have now modified the language in the Discussion section, and it now reads "Very few genes, upon Treg deletion in mice, have shown a severe autoimmune phenotype, similar to that of the Treg-deficiency, including..." (Page 18).

2. It would be good to mention the gender of the mice used. Since the *FoxP3* gene is X-linked, this is important. In fact, this is essential for the interpretation of the RNAseq results, which were performed on (I surmised) female mice, which are heterozygous for the the *FoxP3*-Cre know in allele, such that 50% of their cells are wild type. Although Supplementary figure 5 does explain this, I think this can be made much clearer by mentioning the genders and perhaps adding a sentence in the main body of the manuscript about the X-chromosomal location of the *FoxP3* gene and the fact that female mice are chimeric.

Thanks for this great suggestion. The reviewer was correct that the RNAseq experiments were performed on female chimeric mice. Per reviewer's suggestion, we have now mentioned the genders and added one sentence in the main body of the manuscript as follows: "Note that the *FoxP3* gene is localized in the X-chromosome, and the female mice used here with two genotypes are chimeric" (Page 10). We have also labeled clearly in respective Figure legends when these female chimeric mice were used.

3. It was not clear to me which of the genes/proteins shown in the heat maps in figures 4 and 5 are significantly differentially expressed. This should be clearly indicated. How was the selection of genes shown here made?

Thank you. We should have made it clearer in our original submission. For Fig4a: the known functional molecules of Treg cells were selected, and those with fold change greater than 1.27 were displayed in the heat-map. For the others: KEGG analysis was performed and protein (gene) with changes greater than 2 times and p value less than 0.05 ($p < 0.05$) were selected. The top altered signal pathways by genes were displayed in Supplemental Fig. 12; biologically significant and interesting pathways were selected and displayed in the heat-map. We have added the top altered signal pathways by proteins in this revised vision Supplemental Fig. 13.

4. It is surprising that the Treg-specific *Rbx2* and *Ube2f* deficient mice do not have a Treg phenotype. I suppose the *Rbx2* mice were confirmed previously to lack expression of this protein, (might be good to point that out), but I did not see evidence that the *Ube2f* targeting led to loss of expression.

Both *Sag* (*Rnf7*) and *Ube2f* are expressed in the Treg cells (Supplemental Fig. 25, also see below), which was confirmed by qPCR, western blot or single cell RNA sequencing. These data have been included (Supplemental Fig. 1a & Supplemental Fig. 15a), and discussed in the text (Page 6, 14 & 18).

As for the *Ube2m* deficient mice: is there a possibility that a truncated protein is still made, or perhaps that a splice variant is expressed that lacks the targeted exons, but still has partial function? If that is the case, it will alter the interpretation of the different phenotypes of the *Rbx1* and the *Ube2m* deficiencies. In general, the characterization of the mice is a little thin. Only mRNA expression levels are shown, which are not necessarily a good reflection of functional gene deficiency. Do the RT-PCR primers used for analysis anneal in the exons between the *LoxP* sites? If so, alternative transcripts may still be made. Minimally, it would be helpful to mention the sequences of the primers and to mention where they anneal. In addition, it might be a good idea to test expression of the proteins (this seems especially relevant for the *Ube2m* and *Ube2f* proteins), ideally with antibodies to the N- and to the C-termini of the proteins.

The CRISPR/Cas9-based cKO strategy is to conditionally delete the exons 2-4 with exon 5 frame-shifted in *Ube2m* genome. Given exon 1 only encodes 36 amino acids, there is no possibility that a truncated functional protein is still being made (we have now provided these detailed in the M&M section, Page 25). We have tested *Ube2m* cKO MEF using western blot and found *Ube2m* protein was completely eliminated upon infection with Ad-Cre virus, as compared to infection with Ad-GFP control virus (see below, unpublished data). The same *Ube2m^{fl/fl}* mice were crossed with LyzM-Cre and Ad-SPC-Cre (PMID: 33720974, and manuscript under preparation), and elimination of *Ube2m* mRNA and proteins were fully characterized and confirmed by both RT-PCR and western blot (PMID: 33720974, and data not shown).

The largely elimination of *Ube2m* mRNA in Treg cells from *Fxop3^{cre};Ube2m^{fl/fl}* mice, as compared to the wild-type control mice, was confirmed by both transcriptome analysis and q-PCR, and has been added in the revision (Supplemental Fig. 16a).

Reviewer #2 (Remarks to the Author):

Wu et al

Requirement of Rbx1 Cullin-Ring ligase E3 and Ube2m neddylation E2 in the maintenance of the fitness of Treg cells

Here the authors investigate the effect of a Treg-specific deletion of the Tbx1 component of cullin-RING ligases. This results in early onset fetal inflammatory disorders, disrupted homeostasis and impaired Treg functions. Similar phenotypes were seen in mice with Ube2m, a neddylation E2 ligase in Treg cells but with less severity. They propose that the Ube2-Rbx1 axis is required for the maintenance and homeostasis of Treg cells. They further propose that pevonedistat, a potent inhibitor of neddylation/CRLs, may have therapeutic value.

We thank the reviewer for his/her concise and excellent summary of our work.

General

The CRL family is large. Is there any specificity in expression of components in Treg cells versus other effector populations? It would be interesting, and therapeutically important, to know if Ube2-Rbx1 is Treg cell specific.

A very insightful question indeed. Given this study is mainly focused on Ube2m and Rbx1, we have performed extensive online search and did not find any source to show that the expression of Ube2m-Rbx1 is Treg cell specific. Rather, one study, using sc-RNA sequencing approach, reported a widespread expression of Rbx1 and Ube2m in multiple mouse organs (Han et al. Cell. 2018 172: 1091-1107), as shown below. This paper is now cited in our Discussion section (Page 21).

Despite Rbx1 does not appear to be selectively expressed in Treg cells, it is noting worthy that the biochemical mechanism mediated by Rbx1 in Treg cells may have specificity, at least to some extent. Many common substrates of Rbx1/CRLs1-4, like Cdkn1b/p27 and Foxo1, were not accumulated (Fig. 5d), whereas the others may more

selectively accumulated in Rbx1-deficient Treg cells (Fig. 5a&d). Thus, it appears that Rbx1 substrates are specifically subjected to Rbx1 modulation in Treg cells to maintain their proper functions. (Fig. 5d & Page 13-14). Thus, the biochemical mechanism(s) performed by Rbx1 in Treg cells could be different from those seen in other cells or tissues, an interesting subject for future investigation.

The authors should demonstrate Rbx1- and Ube2- deletion in Treg cells by an alternative approach than relative levels of mRNA by sequencing. Since this is the focus of the whole paper, it would be appropriate here to show both genomic deletion (qRT-PCR) and western blot analysis.

Agreed. We have now provided qRT-PCR data to confirm the deletion of *Rbx1* and *Ube2m* in the revision (Supplemental Fig. 2a & Supplemental Fig. 16a). Unfortunately, we can only obtain very limited number of Rbx1-null Treg cells due to significantly reduced proliferation and increased apoptosis. Thus, it will be technically difficult to perform Western blot.

We indeed performed Western blot for Ube2m. We were able to detect Ube2m in wt, but not in Ube2m-deficient Treg cells (shown below for reviewer's view). However, due to high background of blot, we do not feel comfortable to include it in the manuscript. We are currently harvesting more Treg cells via FACS sorting from Ube2m deficient mice to obtain optimal western result. Thank you.

In figures where ratios and percentages of immune cells are shown, there should also be information concerning absolute numbers to get a better idea of the effects of Rbx1-deletion on immune homeostasis.

The suggestion is well taken, and the absolute numbers of immune cells under various conditions have now been shown in our revision. (Fig. 2e, Fig. 7a, Supplementary Fig. 3 & Supplementary Fig. 17b-d)

I struggle somewhat to understand the specificity and function of Rbx1 in Treg cells. There are dramatic transcriptional changes occurring after deletion of Rbx1, and also proteomic changes. This seems to affect pretty much every major pathway that could be envisaged. Is this not simply because the cells are dying (evident from greatly reduced numbers, reduced proliferation and increased apoptosis)? This really needs to be discussed in detail and some mechanistic insight presented. The study is in general very descriptive and speculative, leaning on a lot of transcriptome analyses, and

mechanistically weak. Is the response of Treg cells to loss of Rbx1 not just a response to induction of cell death in the context of an immune system? How is Rbx1-deletion so dramatically affecting the transcriptome and proteome?

The reviewer was correct that Treg deletion of Rbx1 caused dramatic changes in both transcription levels and proteomic levels, most likely due to its dual E3 activities in promoting ubiquitylation and neddylation (also see our response to Reviewer 1/point #2). These changes cannot be mainly or solely due to reduced proliferation and increased apoptosis, since both transcriptome and proteomic analyses did not indicate so. We agreed that the mechanistic study is rather limited largely due to technical difficulty in obtaining sufficient Rbx1-deficient Treg cells. In fact, we spent nearly one year in mouse mating to get barely enough Treg cells for the mass spectrometric experiment. To make things worse, Treg cells are very difficult to culture *in vitro* and no established Treg line is available for mechanistic study, not to mention that the Rbx1-deficient Treg cells are apoptosis-prone. Thus, it is technically impossible to conduct in-depth mechanistic study by using Rbx1-deficient Treg cell lines. Nevertheless, through both transcriptome and proteome analyses, we did reveal the signaling networks governed by Rbx1 and some possible Rbx1 substrates, specific to Treg cells.

More specifically, please see our response below to your comment on Figure 2. Overall, in this vision, we added additional data to demonstrate that the response of Treg cells to the *Rbx1* loss is the combination of the loss of immune suppressive function and the loss of cellular compartment.

Specific

Figure 1. It would be appropriate to not only provide information concerning CD4+ and CD8+ cells, but also B cells (CD19), DCs (CD11c), macrophages (CD11b) etc to enable a proper understanding of the effect of Rbx1 deletion on immune homeostasis.

Fully agreed. We have included these analyses in our revision, as suggested by the reviewer. (Supplementary Fig. 4 for *Rbx1* deletion and, Supplementary Fig. 18 for *Ube2m* deletion)

Figure 2. The authors conclude these experiments by stating that Rbx1 is needed for the suppressive function of Treg cells. However, they show that Rbx1-deletion leads to greatly reduced numbers of Treg cells due to reduced proliferation (Fig 2b) and increased apoptosis (Fig 2c). I find it difficult to conclude that it is Treg-suppressive function that has been affected. To confirm this they should perform an in vitro or in vivo suppression assay.

Indeed, the *in vivo* suppression assay of Treg cells is the gold standard to determine whether Treg cells in question indeed have the suppressive function. We have now performed this suppression assay. The naive T cells were injected into immune-deficient

Rag1^{-/-} mice in combination with *wt* vs. *Rbx1*-deficient Treg cells. We found that *Rbx1*-deficient Treg cells cannot prevent the occurring of colitis induced by naive CD4⁺-T cells, demonstrating the loss of suppression activity. This newly generated data are included in Fig. 2a.

In addition, the following two lines of evidence also suggest that *Rbx1* is required for the suppressive function of Treg cells:

(1) Robust inflammatory responses in 8-days-old *Foxp3*^{Cre};*Rbx1*^{fl/fl} mice with normal homeostasis of Treg cells (Fig. 2b-e). While the percentage of Treg/CD4⁺-T-cells in the *Foxp3*^{Cre};*Rbx1*^{fl/fl} mice decreased gradually after the birth, the Treg/CD4⁺ ratio is normal in *Foxp3*^{Cre};*Rbx1*^{fl/fl} mice at the age of 8 days (Fig. 2b), and the numbers of lymphocytes and Treg cells are also similar, as compared to that of the wild type control mice (Fig. 2c&d). However, these 8 days-old *Foxp3*^{Cre};*Rbx1*^{fl/fl} mice have already developed robust over-activated immune response (Fig. 2e). The only possible explanation is that impaired suppressive function of *Rbx1*-deficient Treg cells leads to such obvious immune over-activation. We have now discussed this important point in the Discussion section (Page 18-19).

(2) Th1/Th2-dominated immune-activation in *Foxp3*^{Cre};*Rbx1*^{fl/fl} mice. If the phenotype of *Foxp3*^{Cre};*Rbx1*^{fl/fl} mice were simply the consequence of the decreased homeostasis of Treg cells, one would expect that all types of inflammatory reactions are activated, without any specificity. However, in *Foxp3*^{Cre};*Rbx1*^{fl/fl} mice, the Th1/Th2-related cytokines and antibodies were elevated robustly, while the Th17-related cytokines and antibodies were only modestly changed (Supplementary Fig. 5& Supplementary Fig. 6). Thus, the inflammatory response in *Foxp3*^{Cre};*Rbx1*^{fl/fl} mice is Th1/Th2-dominated, indicating *Rbx1* controls the suppressive function against Th1/Th2-mediated inflammation. Namely, *Rbx1* plays specific function in Treg cells, rather than simply controls proliferation and survival. We have included this point in the Discussion section as well in this revision. (Page 18-19)

Figure 3. The authors should run some direct transcriptome comparisons between clusters 0, 3 and 4 and online (effector) Treg datasets rather than handpicking some Treg-related genes. How similar are these clusters to predefined PB or tissue-resident Treg cell transcriptomes? To conclude that “Rbx1 is essential for maintenance of effector subpopulations of Treg cells” it would be relevant to confirm this using FACS-based technologies and not just sc-RNA-seq.

The reviewer’s point is well taken. It is indeed a great idea to make such a comparison. However, the single-cell transcriptome sequencing analysis was generally performed on whole blood samples or single cells derived from digested tissues. The samples contained many types of cells with very small fraction of Treg cells, and also Treg cells were often only being categorized as one whole group of T cells without further dividing into subpopulations.

In this study, we sorted out Treg cells and took a large number (~10,000) for single cell sequencing analysis, and then subdivided them into 13 subpopulations, based upon

transcriptome profiling. So far only one such a study was reported recently (Miragaia, R. J. et al. *Immunity* 2019, PMID: 30737144). Our sequencing results are in general consistent with this previous report, but also with some differences. It is worth noting that in this previous study (Miragaia, R. J. et al. *Immunity* 2019, PMID: 30737144), the authors also compared Treg cells with peripheral lymphocytes from different tissues of mice via single-cell sequencing, and found a very different expression profiles, showing different cell populations in the t-SNE map, which did not overlap at all. We have now discussed it in the Discussion section (Page 20).

Per reviewer's suggestion, we have now performed FACS-based profiling to confirm the transcriptome data. As shown in Fig. 3a&b, the single-cell transcriptome sequencing showed that in *Rbx1*-deficient Treg cells, percentage of Clusters 3 and 4 were decreased. Among the marker genes represented in Fig. 3c, high level of CD44 and low level of Sell (CD62L) is an obvious character of Clusters 3 and 4. We have now confirmed this finding by the FACS analysis by showing a ~3-4-fold reduction of CD44^{hi}CD62^{lo} cells in Treg population (Page 10). This newly generated data is now included in Fig. 3d to support our conclusion that "*Rbx1* is essential for the conversion of quiescent to effector subpopulations of Treg cells".

Figure 4. In Figure 4a-b the authors list a set of important genes for Treg cell function whose expression is changed by loss of Rbx1. It seems that very few of these actually demonstrate changes on the protein level (as evidenced by the mass spec data). What is the relevance of the data presented in Figure 4b if the protein level of these genes is not affected by Rbx1-deletion? If the authors wish to propose that Rbx1 is involved in energy and nucleotide metabolism then they should provide some experimental data to support this.

We thank the reviewer for this very reasonable question. The limitation of our study is the technical difficulty in obtaining sufficient number of *Rbx1*-deficient Treg cells for Mass Spect analysis (described above as well). With these limited numbers of cells, we were only able to detect a little over 3000 proteins by mass spectrometry, which is much lesser than the numbers of genes detected by transcriptome analysis. Thus, the purpose of Figure 4b is to list possible transcriptional changes upon *Rbx1* depletion in Treg cells to demonstrate the robust effect of *Rbx1*-deficient in gene expression. We have modified the paragraph, and it now reads: "The results suggest that *Rbx1* controls the expression of numerous key genes regulating the processes of inflammation, immunological responsiveness, proliferation/survival, and metabolisms, which is subjected to further experimental validation" (Page 11).

We also agreed with the reviewer's comment that "*If the authors wish to propose that Rbx1 is involved in energy and nucleotide metabolism then they should provide some experimental data to support this*". Given the technical limitation, we were unable to confirm the proteomics data by Western blotting. We, therefore, softened the language and only mention the altered levels of these proteins (via proteomics analysis) and their known involvement in energy and nucleotide metabolism and add one sentence in the end

of that paragraph, it reads “which is again subjected to further experimental validation” (Page 12).

Figure 6. As also mentioned above, the authors should do a proper characterization of immune cell types and numbers here, and also compare this to Rbx1-deletion. Just ratios of Treg/CD4 are not sufficient to understand the phenotype. Can the authors explain the increased number of Treg cells in the LN? Are the Ube2m-deleted Treg cells able to suppress effector proliferation in vitro or in vivo?

Thank you. In this revision, we have characterized these parameters (Fig. 7a, right panel & Supplementary Fig. 17b-d for Ube2m). The data for Rbx1-deletion were shown in Supplementary Fig. 7b and Supplementary Fig. 3, respectively. Please note that the age of *wt* mice used for comparison was 19 days for Rbx1 and 4 months for Ube2m. So actually number of Treg cells are not the same.

In response to inflammation, the immuno-suppressive Treg cells normally proliferate to inhibit the inflammation and restore the immune homeostasis of the body (PMID 23135404, and 23812589). Thus, the ratio and number of Treg cells are often elevated *in vivo*, if the proliferation and survival of Treg cells are not inhibited, which is the case for Ube2m-deficient Treg cells. We have now provided this explanation in the text (Page 15).

Per reviewer’s request, we used *Rag1*^{-/-} mice to determine whether the *in vivo* suppressive function of Ube2m-deficient Treg cells are impaired. Briefly, the naive CD4⁺ T cells were injected into immune-deficient *Rag1*^{-/-} mice in combination with *wt* vs. Ube2m-deficient Treg cells. Unlike *wt* Treg cells, the Ube2m-deficient Treg cells cannot prevent/inhibit the colitis, induced by the naive CD4⁺ T cells. We, therefore, concluded that the suppressive function of Ube2m-deficient Treg cells is indeed impaired. (Fig. 7d; Page 15-16)

Minor

The introductory paragraph does not really give justice to the field. It suggests there are a small number of genes with essential roles in Treg biology and that “studies are rather limited as to how Treg cells are precisely regulated”. There is in fact a body of literature reporting studies investigating Treg cell biology. I think the authors could better just focus on a lack of information concerning CRLs in Treg cell biology rather than this general statement.

The comment is well taken. To avoid misleading, we will emphasize the lack of information concerning protein neddylation and CRLs in Treg cell biology, as suggested by the reviewer. It now reads “There is no systematic study using mouse knockout models to elucidate physiological role of CRLs (via Rbx1/Rbx2) and neddylation (via Ube2m/Ube2f) in regulation of functions and survival of Treg cells for proper controlling of body’s immune system”. (Page 4-5). We also rearranged the Introduction section by introducing neddylation-CRLs first.

We agreed with reviewer's statement that "*There is in fact a body of literature reporting studies investigating Treg cell biology*". We have partially addressed this in our response to reviewer 1 (point #1), and we should have made our statement clearer in our original submission. As a matter of fact, since 2003 when Foxp3 was identified as a specific marker of Treg cells, many investigators in the field had used the Foxp3^{Cre}-loxp system to selectively knock out the genes of their interest in mouse Treg cells in an attempt to find other key regulators. In general, if Treg deletion of a given gene reproduces the phenotype of Treg cell depletion, namely the mice develop a severe autoimmune response and die around 30 days, the gene in question is critically essential for Treg cells. However, only a handful of genes meet this stringent criteria, including *Foxp3* (PMID: 12522256/ 12612578/12612581), *Foxo1* (PMID: 23135404), *Raptor* (PMID: 23812589), *Lkb1* (PMID: 28847007), *Uqcrsfl* (PMID: 30626970), and *Pggt1b* (PMID: 33207246). Our study identified *Rbx1* as a new addition to this limited list.

The reviewer was correct that many other genes are also important and Treg deletion of these genes indeed caused inflammatory response, but to lesser extent with mice survived 70 days or beyond. We have now modified the language in the Discussion section, and it now reads "Very few genes, upon Treg deletion in mice, have shown a severe autoimmune phenotype, similar to that of the Treg-deficiency, including..." (Page 18).

Figure 5b. Presenting one FACS blot is not sufficient, there should also be quantification of multiple experiments.

Agreed. We have repeated few times of FACS blot and now included quantified data in this revision (Fig. 5b, bottom panel).

The authors should remove the discussion of pevonedistat from the abstract and introduction as they do not perform any experiments showing that this effects Treg cell function. They state in the discussion that "our study provides a proof-of-concept evidence that pevonedistant may have novel application in the treatment of diseases associated with over-activation of Treg cells". I don't agree with them that they have really provided proof-of-concept and this should be rephrased. It would certainly be interesting, and also improve the relevance of the manuscript, if they could perform in vivo, or minimally in vitro experiments to validate this hypothesis.

The comments are well taken. Per reviewer's suggestion, we have removed the discussion of pevonedistat from the sections of Abstract and Introduction. In the Discussion section, we have rephrased our previous statement and now it reads "Given that inactivation of either neddylation E2/Ube2m, or E3/Rbx1 caused suppression of Treg function, MLN4924, a small molecule inhibitor of neddylation, currently in several Phase II clinical trials for anticancer application, may have a new application in the treatment of disease associated with over-activation of Treg cells, which is certainly worth investigating experimentally." (Page 23)

Reviewer #3 (Remarks to the Author):

The authors build on their earlier studies Rbx1 and Sag2/Rbx2 whole animal knockouts. Here they delete Rbx1, Sag2/Rbx2, Ube2m, or Ube1 in Tregs using the Foxp3cre driver. These genes encode critical components of CRL E3 multi-unit ubiquitin ligases and neddylation regulatory pathway proteins. UBE2M with RBX1 serve in neddylation of CUL1-CUL4. UBE2F with SAG2/RBX2 serve in neddylation of CUL5. The authors perform several standard assays in the field to examine effects on animal physiology and proliferation, viability, and function of Tregs in the knockouts. The deletion of SAG2/RBX2 and UBE2F have no obvious effects. The deletion of RBX1 is reminiscent of loss of Foxp3 and generally reflects RBX1 as essential for Tregs in vivo. The deletion of Ube2m has an intriguing phenotype. The mice suffer from inflammatory disorders, but not with the same severity, rate, or penetrance of Rbx1.

For molecular analyses, the authors perform scRNA-seq and proteomics on the Rbx1 null Tregs, with some western blots, and transcriptome analyses on the Ube2m nulls. They conclude that Rbx1 plays Ube2m-dependent and -independent roles in Treg cells.

We thank the reviewer for his/her excellent summary of our work.

From an immunological perspective, the results do not seem surprising and scale with the known essentiality of these genes for proliferation.

In our responses to first two reviewers, particularly Reviewer #2, Figure 2, we have addressed this concern and concluded that the role of Rbx1 in proliferation cannot fully explain our experimental observations, since 1) Robust inflammatory responses in 8-days-old *Foxp3^{Cre};Rbx1^{fl/fl}* mice with normal homeostasis of Treg cells; 2) T_H1/T_H2-dominated immune-activation in *Foxp3^{Cre};Rbx1^{fl/fl}* mice. In case of Ube2m, we found Ube2m actually regulates Treg cell function independent of proliferation and survival (Fig. 7).

Furthermore, per Reviewer 2's suggestion, we have performed an *in vivo* suppression assay and found that Treg cells deficient of either Rbx1 or Ube2m have impaired suppressive function (Fig. 2a and Fig. 7d). We have now added one paragraph in the Discussion section to fully discuss this important point (Page 18-19).

Major comments:

1. In terms of the experiments, the authors should do the control experiments for Rbx1 and Ube2m and transfer Tregs from WT mice into the newborn mutants to be sure they rescue the immune phenotypes.

Thank you for this excellent point!

I transferred Treg cells from WT mice into the *Foxp3^{Cre};Ube2m^{fl/fl}* mice, and found that wt Treg cells indeed rescue the immune phenotypes. (Fig. 7e&f, Page 16)

We also transferred Tregs from WT mice into *Foxp3^{Cre};Rbx1^{fl/fl}* mice at p13, and

found that wt Treg cells cannot rescue the immune phenotypes. We repeated the experiment using *Foxp3^{cre};Rbx1^{fl/fl}* mice at p8 (at the stage with normal Treg homeostasis, Fig. 2b&c) as the recipient, and the results showed that wt Treg cells still cannot rescue the immune phenotypes (Fig. 2f&g). We have provided possible explanation on this phenomenon (Page 8-9).

2. At this point, there have been so many high throughput CRISPR screens defining essential gene sets and the effects on Tregs scale well with the predicted essentiality of the genes. Rbx1 has been defined as amongst the most essential genes for cell proliferation and the phenotype may well reflect this instead of an effect on specific substrates. In this regard, it is not clear that the proteomics adds the value implied for identifying substrates of Rbx1 E3 ligase with a Treg specific function but more reflects essentiality of Rbx1. Without substantial additional effort at validation, it is not clear what the proteome data provide in terms of mechanism. The conclusion that the proteomic differences are true substrates should be toned down.

We completely agreed with the reviewer that Rbx1 is among the most essential genes for cell proliferation. However, the function of Rbx1 is executed through its E3 ligase activity by directly promoting ubiquitylation and degradation of its variety of substrates. Specifically, Rbx1 is the RING component of CRLs, which is the largest family of the E3 ubiquitin ligases that control the ubiquitylation and degradation of ~20% cellular protein (Soucy et al, Nature, 2009, PMID: 19360080). It is fully expected that single substrate of Rbx1 will not be responsible for robust phenotypes observed. The purpose of this proteomics analysis is to determine, after comparing with transcriptome data, whether we are able to identify possible Rbx1 substrates specifically for Treg cells (for those only accumulated at the protein levels, but not at the mRNA levels upon Rbx1 depletion). It appears that we have achieved this goal by identifying a list of candidates which has not been previously reported in other cellular systems. Unfortunately, confirmation of this, however, is technically challenging, since there is no established Treg cell lines, and mice with Rbx1-deficient Treg cells died very early. Nevertheless, we are currently following up one lead, proapoptotic protein Bim as Rbx1 substrate in Treg cells. Finally, in this revision we have toned down the conclusion in the proteomic difference, per reviewer's suggestion, and add one sentence in the end of that paragraph, in the Discussion section, which reads "Future studies are geared to validate these candidates identified in this study and to characterize their respective roles in functional regulation of Treg cells". (Page 22)

It is worth noting that the proteomics data adds the Treg-specificity of Rbx1 in its mechanism, which is valuable. Many common substrates of Rbx1/CRLs1-4, like Cdkn1b/p27 and Foxo1, were not accumulated in Rbx1-deficient Treg cells (Fig. 5d). Thus, it appears that Rbx1 substrates are specifically subjected to Rbx1 modulation in Treg cells to maintain their proper functions. (Fig. 5d & Page 13-14). thus, the biochemical mechanism(s) performed by Rbx1 in Treg cells could be different from those seen in other cells or tissues.

3. *The interesting results come from the comparison to Ube2m. It is unfortunate for the authors that some people might expect Ube2m would have milder effects because Ube2f technically could compensate. But this reviewer thinks enough investigators would not have this depth of knowledge and would find this intriguing, if the mechanisms were clarified. Some mechanisms that come to mind would be if Ube2f is compensating. The authors must have made the double null already and this could be tested. Or if neddylation is provided by unknown enzymes, this could be tested by pharmacologic or genetic ablation of Nedd8 E1 activity. Another possibility is that neddylation is not so essential in Tregs, a result that could also come from the latter experiments.*

As an expert in the CRL field, the reviewer provided very insightful explanation to our unexpected results. The reviewer was correct that we indeed generated mice with Treg double knockout of Ube2m and Ube2f, and our preliminary analysis revealed that double KO mice have much severe phenotype than Ube2m alone, indicating that Ube2f indeed compensates some functions of Ube2m in Treg cells. Furthermore, the phenotype severity in *Foxp3^{Cre};Ube2m^{fl/fl};Ube2f^{fl/fl}* mice was still less than that of *Foxp3^{Cre};Rbx1^{fl/fl}* mice. Unlike Rbx1-deficient Treg cells, the *Ube2m/2f* double knockout Treg cells did not undergo obvious proliferation arrest and apoptosis (unpublished observations, not included in this study). Taken together, Rbx1 does have special function independent of neddylation in Treg cells, which is a key finding of our study (we stated this in the Discussion section, Page 18). It will be very interesting in determining to what extent that the phenotype severity in mice is attributable to genetic ablation of Nedd8 E1 in Treg cells, which is, however, out of the scope of this manuscript.

4. *The differences between the effects of deleting Rbx1 and Ube2m require more explanation. The Treg/CD4⁺ ratio for Rbx1 make sense but for Ube2m the ratio is increased compared to WT controls. This is counterintuitive and the authors should provide a mechanism.*

In response to inflammation insult, the immuno-suppressive Treg cells usually proliferate to inhibit the inflammation and restore the immune homeostasis of the body (PMID 23135404, and 23812589). The ratio (Treg/CD4⁺) and number of Treg cells are, therefore, elevated *in vivo* under inflammation, if the proliferation and survival of Treg cells are not inhibited, which is the case for Ube2m-deficient Treg cells, but not for Rbx1-deficient Treg cells. The later suffers from reduced proliferation and elevated apoptosis, leading to a decreased Treg/CD4⁺ ratio. We have now included an explanation on this observation in the text (Page 15).

Minor points:

1. *This is not entirely the common view in the field and the authors should either remove this sentence or provide a more balanced perspective:*

RBX1 binds to CDC34/UBCH2C to promote ubiquitylation of large number of substrates

via the K48 linkage, whereas RBX2 binds to UBE2C/UBE2S to promote ubiquitylation of different set of substrates via the K11 linkage for proteasome degradation^{17,18}.

Per reviewer's suggestion, we have now toned down this statement. It reads now "our recent studies showed that RBX1 binds to CDC34/UBCH2C to promote ubiquitylation of large number of substrates via the K48 linkage, whereas RBX2 binds to UBE2C/UBE2S to promote ubiquitylation of different set of substrates via the K11 linkage for proteasome degradation, demonstrating a new layer of difference between two family members". (Page 3-4)

2. *Transcriptome is mis-spelled in figures.*

Thanks, and this have been corrected in our revision.

3. *The authors may wish to elaborate further on the implications of the Rbx2 and Ube2f nulls for Treg regulation by SOCS proteins.*

Thanks, we have done it now, as shown in the last sentence in the Discussion section: "Lack of phenotype by deletion of *Rbx2/Sag* or *Ube2f* in Treg cells excludes the possible involvement of CRL5 in Treg regulation, although the family of SOCS proteins, the known components of CRL5, have immune suppression function (with citation of these two review papers: PMID32334051 and 31898233). (Page 24)

Finally, we would like to take this opportunity to thank all three thoughtful, insightful and critical reviewers for their constructive critiques to our manuscript. We believe that our revision has been significantly strengthened after incorporation of our responses to their comments.

Thank you very much for your consideration and I look forward to hearing from you in near future.

With best regards,

Yi Sun, M.D., Ph.D
Qiushi Chair Professor
Institute of Translational Medicine
Zhejiang University, P.R. China

REVIEWER COMMENTS

Reviewer #1 (Remarks to the Author):

The authors made a commendable effort to try to address my comments and succeeded for some. Still, I think I am not much closer to understanding the role of Rbx1 (and Ube2m) in the biology of Tregs. The major function seems to lie during the transition of naive into effector Tregs. The authors argue that Rbx1 likely controls transcription factors involved in the transition, but they do not provide the data to support this idea. It seems to me that there is a simpler model, for which they actually do have some evidence. The identification of Bim as a substrate would fit with a role for Rbx1 in controlling the survival of the cells after activation/during differentiation into effector Treg cells. One wonders whether the phenotype of the mice could be rescued by elimination of Bim. The authors seem to reject this model, but I am not sure that their arguments are sufficiently strong.

First, I do not think that the new experiments (adoptive transfer of Rbx1 deficient Tregs into Rag knock out mice, adoptive transfer of wild type cells into Rbx1fl/FoxP3-Cre mice) disprove a model based on cell death. To protect Rag1 ko mice from attack by infused conventional T cells, the transferred Rbx1 deficient Tregs must also differentiate into effector Treg cells, and if they die during this transition, this would explain the lack of control. I am not entirely sure what the goal of the other experiment (wild type Tregs into Rbx1fl/FoxP3-Cre mice) was. However, it seems to me that the most logical explanation for the failure to rescue these mice lies in the fact that the wild type Tregs were infused after inflammation had already started. What this teaches us is not clear to me.

Another argument used by the authors against survival as the main mechanism is that young mice develop pathology, even though their Treg numbers are similar to those in wild type mice. However, differentiation of effector Tregs is necessary also to protect these young mice from developing inflammatory disease. What do the effector Treg populations look like in these mice? Do Rbx1 deficient effector Tregs appear in the tissues (this could be examined in female mice)?

The authors also argue against cell death as a mechanism, because there is specific dysregulation of Th1 and Th2 responses, but not of Th17 responses in mice with Treg specific Rbx1 deficiency. However, I believe that the consequence of Treg loss of function is highly impacted by the specific conditions of the mouse facility, with a major influence of microbiota. In other words, perhaps the mice are just not that prone to generating Th17 responses, in which case the argument of selectivity loses its strength.

So in conclusion, I am on the fence about this manuscript. On the one hand, the results are striking, the manuscript is experimentally sound and may be of interest to people in the Treg field and the Ubiquitination/ubidylation field. On the other hand, there are weaknesses to me:

1. the rationale for examining Rbx1 in Tregs seems a little arbitrary.
2. it is not clear in what process the function of Rbx1 is mobilised and how that happens
3. The manuscript does not identify a real mechanism.

While not all three weaknesses have to be fully covered and I realise the authors have already done a tremendous amount of work, in my view at least one of these weaknesses should be addressed more definitively.

Reviewer #2 (Remarks to the Author):

While the study does remain descriptive and still lacks mechanistic insight, the amount of effort the authors have put into revising the manuscript is certainly appreciated. In terms of my original comments the authors have done a good job in addressing most of the issues raised.

1. The authors have analyzed an available online scRNA-seq data set and found no real Treg specificity for Ube2m-Rbx1. This suggests that they will not likely be good therapeutic target.

There does seem to be some Treg selective protein accumulation in Rbc1-deficient Treg cells but the mechanism underlying this is unclear. Understanding this mechanism is indeed beyond the scope of the study. They have briefly discussed this in the discussion.

2. The authors have now demonstrated by qRT-PCR that there are greatly reduced Rbx1 and Ube2 mRNA levels. I understand the problem of performing Western Blotting on these Treg cells.

3. The authors have now included absolute numbers of immune cells improving interpretation of their data.

4. They have not been able to provide any further mechanistic insight into the action of Rbx1 in Treg cells. I understand that this is technically challenging. I remain concerned about the broad and dramatic transcriptional and proteomic changes are more a result of loss of an essential gene rather than specifically reflecting Treg-specific Rbx1 action. Unless the authors can really provide data to show this is not the case, I think they should raise this point in the discussion.

Figure 1. The authors have now included this additional information. There are increased numbers of many immune cell types in Rbx1 and Ube2 knockouts.

Figure 2. The authors have performed in vivo suppression assays to indeed show that Rbx1 is needed for Treg suppressive function. However, the quantification of this data seems to be missing and should be included eg colitis score

Figure 3. I still believe it is valid to compare Treg transcriptomes from their clusters with online data of tissue-resident Treg cells. However, I am satisfied with the authors rebuttal.

Figure 4. The authors have modified the text but over-state the data. On page 11 they should rephrase the sentence to say " The results suggest that loss of Rbx1 results in changes in the expression of...". They do not show that Rbx1 directly controls gene expression and the study lacks mechanistic insight into these events.

Figure 6. The authors have now made the analyses requested.

Discussion. The discussion is rather speculative in places and could be reduced in length. For example, as mentioned before, the authors provide no evidence that metabolic pathways are affected, so it doesn't make sense to me to highlight this relative to many other targets that are also regulated. A minor point: on page 21 they state "Nevertheless, our work presented the first proteomic analysis on mouse Treg cells". This sentence is incorrect and should be deleted. See for example: PMID: 28373295. Also I would move up the last paragraph of the discussion, it feels out of place.

All other Minor Comments have been addressed.

Reviewer #3 (Remarks to the Author):

The authors added important experiments to address reviewers, particularly the controls needed to confirm that the phenotypes result from loss of Rbx1. The data, as well as the mouse models, will be of great utility to the field. As such, I recommend publication in Nature Communications with the condition that the data (not only RNA-seq and microarray but also proteomics) and mice be made available to others upon publication. The authors should consider the following minor point when preparing the final version of their manuscript:

The introduction could still provide a more balanced view of the biochemical mechanisms but as the study largely focuses on immune functions I leave this to the authors' discretion.

Responses to the reviewers

Reviewer #1 (Remarks to the Author):

The authors made a commendable effort to try to address my comments and succeeded for some. Still, I think I am not much closer to understanding the role of Rbx1 (and Ube2m) in the biology of Tregs. The major function seems to lie during the transition of naive into effector Tregs. The authors argue that Rbx1 likely controls transcription factors involved in the transition, but they do not provide the data to support this idea. It seems to me that there is a simpler model, for which they actually do have some evidence. The identification of Bim as a substrate would fit with a role for Rbx1 in controlling the survival of the cells after activation/during differentiation into effector Treg cells. One wonders whether the phenotype of the mice could be rescued by elimination of Bim. The authors seem to reject this model, but I am not sure that their arguments are sufficiently strong.

Thanks for your insightful comments. We have now performed a *Bim* rescue experiment by simultaneous deletion of *Rbx1* and *Bim*. The mice with Treg deletion of both *Rbx1* and *Bim* also suffered an early-onset fetal inflammatory disorders with a similar survival rate as Treg deletion of *Rbx1* alone. However, the decreased CD4⁺/CD8⁺ T-cell ratio and increased proportion of effector/memory T cells (CD44^{hi}CD62L^{lo}) among populations of Tcon cells have been significantly improved upon *Bim* deletion, even although these parameters are still significantly altered when compared to wild-type mice. We, therefore, concluded that *Bim* plays a relatively minor effect on Treg *Rbx1*-deficient phenotypes. These newly generated data are now included in Fig. 6 and supplemental Fig. 15 (pp 15-16, highlighted in green).

First, I do not think that the new experiments (adoptive transfer of Rbx1 deficient Tregs into Rag knock out mice, adoptive transfer of wild type cells into Rbx1fl/FoxP3-Cre mice) disprove a model based on cell death. To protect Rag1 ko mice from attack by infused conventional T cells, the transferred Rbx1 deficient Tregs must also differentiate into effector Treg cells, and if they die during this transition, this would explain the lack of control. I am not entirely sure what the goal of the other experiment (wild type Tregs into Rbx1fl/FoxP3-Cre mice) was. However, it seems to me that the most logical explanation for the failure to rescue these mice lies in the fact that the wild type Tregs were infused after inflammation had already started. What this teaches us is not clear to me.

1. Thanks! We value your thoughtful comments. The *in vivo* suppression assay is a general method to measure the suppressive function of Treg cells. While these two adoptive transfer experiments neither support nor “disprove a model based on cell death”. Our newly performed *Bim* rescue experiment, however, excluded the major contribution of *Bim*-induced apoptosis to observed immune defective phenotypes observed in *Rbx1fl/FoxP3-Cre* mice, and supported an immune-suppression model. In addition, we showed that while *Foxp3^{cre};Rbx1^{fl/fl}* mice at the p8 stage have normal levels of Treg/CD4⁺ ratio, the numbers of Treg cells, and total numbers of lymphocytes, they still showed an over-activated immune system with a ~3-4-fold increase in proportion and cell number of effector/memory (CD44^{hi}CD62L^{lo}) T cells (Fig2b-e). Taken together, our conclusion that *Rbx1*-deficiency leads to an impaired suppressive function of Treg cells, is supported by these results.

Furthermore, we believe what we learned from the second transfer experiment (wt Tregs to Rbx1-fl/FoxP3-Cre or Ube2m-fl/FoxP3-Cre mice) is that the inflammatory immune phenotype is reversible for Foxp3-cre;Ube2m^{fl/fl}, but not for Foxp3-cre;Rbx1^{fl/fl} mice. The possible explanation is that the infused wt Treg cells did not have enough time window for functional rescue after inflammation had already started in the recipient infant Foxp3^{cre};Rbx1^{fl/fl} mice (wt Treg cells were infused at the p8 stage and mice are dying at the p20 stage). We have now fully discussed these points (Pages 10&18, highlighted in green).

Another argument used by the authors against survival as the main mechanism is that young mice develop pathology, even though their Treg numbers are similar to those in wild type mice. However, differentiation of effector Tregs is necessary also to protect these young mice from developing inflammatory disease. What do the effector Treg populations look like in these mice? Do Rbx1 deficient effector Tregs appear in the tissues (this could be examined in female mice)?

Excellent point! Given that CD44 high and CD62L low (CD44^{hi}CD62L^{lo}) can be used to identify the Clusters 3 and 4 effector Treg subpopulations (Fig. 3c, d, f), we performed the FACS analysis to measure the effector Treg populations in peripheral lymph node tissues of infant mice (8-day-old) with the genotype of wt vs. Foxp3-cre;Rbx1^{fl/fl}. The newly generated data showed that proportion of effector subpopulation of Treg cells were indeed decreased in Rbx1-deficient Treg cells at such an early age (Fig. 3e & Supplementary Fig. 11b), with reduced levels similar to what was observed in older mice at age of ~20 days (Fig. 3d & Supplementary Fig. 11a), indicating that the reduction in the Treg effector subpopulations upon Treg Rbx1 deletion is an early event. We have now described the finding in the text (Pages 11-12, highlighted in green).

We also analyzed the subpopulations of Treg cells in the livers of wt vs. Foxp3-cre;Rbx1^{fl/fl} mice (p8) by FACS analysis. Unexpectedly, as shown on the right, the markers used for analysis of Treg cells from lymph nodes (CD44 and CD62L) are able to distinguish CD4⁺Foxp3⁺ Tcon cells (bottom panels), but are unable to distinguish Treg (CD4⁺Foxp3⁺ Tcon) subpopulations in the liver (top panels), indicating a possible difference of Treg subpopulations between lymph nodes and liver.

Indeed, it has been reported that Treg cells from different tissues have distinct transcriptome patterns (Miragaia, R. J. et al. Immunity. PMID: 30737144). Given that we have provided the data from lymph nodes, we believe that Treg distributions in other tissues (e.g. liver) upon Rbx1 deletion is beyond the scope of this manuscript, and hope you will agree with us. Thank you.

The authors also argue against cell death as a mechanism, because there is specific dysregulation of Th1 and Th2 responses, but not of Th17 responses in mice with Treg specific Rbx1 deficiency. However, I believe that the consequence of Treg loss of function is highly impacted by the specific conditions of the mouse facility, with a major influence of microbiota. In other words, perhaps the mice are just not that prone to generating Th17 responses, in which case the argument of selectivity loses its strength.

A valid point! Given we do not have direct evidence as to whether our mouse facility (built in 2018, following the international SPF standard) impacts mouse microbiota, we have now tuned down our language on this Th17 response as a strong argument against cell death mechanism.

So in conclusion, I am on the fence about this manuscript. On the one hand, the results are striking, the manuscript is experimentally sound and may be of interest to people in the Treg field and the Ubiquitination/neddylaton field. On the other hand, there are weaknesses to me:

- 1. the rationale for examining Rbx1 in Tregs seems a little arbitrary.*
- 2. it is not clear in what process the function of Rbx1 is mobilised and how that happens*
- 3. The manuscript does not identify a real mechanism.*

While not all three weaknesses have to be fully covered and I realise the authors have already done a tremendous amount of work, in my view at least one of these weaknesses should be addressed more definitively.

Thank you for your fairness.

We have now provided detailed justification/the rationale for examining Rbx1 in Tregs; Specifically, we have stated that, previous studies have shown that the components of neddylation-CRLs system (particularly several subunits of CRLs) plays essential roles in functional regulation of multiple immune cells, including macrophage, T-lymphocytes, and DCs; however, no systematic study on the role of neddylation-CRLs system in regulation of Treg cell function have been reported. So, our study filled in this scientific gap by investigating systematically the function of neddylation-CRLs system (Ube2m&Ube2f, Rbx1&Sag) in Treg cells (Page 4-5, highlighted in green).

It is indeed difficult to elucidate “in what process the function of Rbx1 is mobilised and how that happens”, given the involvement of Rbx1 in regulation of both neddylation and ubiquitylation.

Our Bim rescue experiment (Figure 6, Supplementary Figure 15) should be considered identifying a real mechanism, which further demonstrated that the function of Rbx1 in Treg cells, is likely mediated by multiple downstream targets, given Rbx1 acts as E3 for both neddylation and ubiquitylation. Finally, the cell-based mechanistic study for Treg cells is restrained due to the lack of available cell culture models.

Reviewer #2 (Remarks to the Author):

While the study does remain descriptive and still lacks mechanistic insight, the amount of effort the authors have put into revising the manuscript is certainly appreciated. In terms of my original comments the authors have done a good job in addressing most of the issues raised.

Thank you for your positive comment on this revision.

1. The authors have analyzed an available online scRNA-seq data set and found no real Treg specificity for Ube2m-Rbx1. This suggests that they will not likely be good therapeutic target. There does seem to be some Treg selective protein accumulation in Rbx1-deficient Treg cells but the mechanism underlying this is unclear. Understanding this mechanism is indeed beyond the scope of the study. They have briefly discussed this in the discussion.

Thank you. We do have now included a Bim rescue experiment to provide at least one solid mechanism (Figure 6, supplementary Figure 15, Page 15-16 highlighted in green).

2. The authors have now demonstrated by qRT-PCR that there are greatly reduced Rbx1 and Ube2m RNA levels. I understand the problem of performing Western Blotting on these Treg cells.

Thank you for your understanding.

3. The authors have now included absolute numbers of immune cells improving interpretation of their data.

Thanks again.

4. They have not been able to provide any further mechanistic insight into the action of Rbx1 in Treg cells. I understand that this is technically challenging. I remain concerned about the broad and dramatic transcriptional and proteomic changes are more a result of loss of an essential gene rather than specifically reflecting Treg-specific Rbx1 action. Unless the authors can really provide data to show this is not the case, I think they should raise this point in the discussion.

We value your comment, and agreed that the robust lethal auto-immune phenotypes, along with dramatic transcriptional and proteomic changes, is a result of loss of an essential gene, like Rbx1. Indeed, our previous study using the total KO mice model showed that Rbx1 is indeed an essential gene for embryonic development, and deletion of a single substrate (p27) only extend embryonic life for 3 days (PNAS, 2009/PMID: 19325126). Moreover, our Bim rescue experiment, showing minor, if any, effect in reversing lethal autoimmune phenotypes, further support the notion. On the other hand, however, we did find in our proteomic study that many common substrates of Rbx1/CRLs1-4, reported in human cancer cell lines, were not accumulated in Rbx1-deficient Treg cells. Thus, it

appears that Rbx1 substrates are rather specifically subjected to Rbx1 modulation in Treg cells to maintain their proper functions, which suggests a Treg-specific Rbx1 action. We have now fully discussed this point in the manuscript (Page 14, highlighted in green).

Figure 1. The authors have now included this additional information. There are increased numbers of many immune cell types in Rbx1 and Ube2 knockouts.

Thank you.

Figure 2. The authors have performed in vivo suppression assays to indeed show that Rbx1 is needed for Treg suppressive function. However, the quantification of this data seems to be missing and should be included eg colitis score

We have now included this missing quantification data (Page 9&18&29, highlighted in green).

Figure 3. I still believe it is valid to compare Treg transcriptomes from their clusters with online data of tissue-resident Treg cells. However, I am satisfied with the authors rebuttal.

Thank you.

Figure 4. The authors have modified the text but over-state the data. On page 11 they should rephrase the sentence to say “The results suggest that loss of Rbx1 results in changes in the expression of...”. They do not show that Rbx1 directly controls gene expression and the study lacks mechanistic insight into these events.

Agreed. Indeed, Rbx1 is not a transcription factor and the expression changes upon Rbx1 loss is rather an indirect effect. We have modified this sentence, per reviewer’s suggestion. (Page 13, highlighted in green)

Figure 6. The authors have now made the analyses requested.

Thank you.

Discussion. The discussion is rather speculative in places and could be reduced in length. For example, as mentioned before, the authors provide no evidence that metabolic pathways are affected, so it doesn’t make sense to me to highlight this relative to many other targets that are also regulated. A minor point: on page 21 they state “Nevertheless, our work presented the first proteomic analysis on mouse Treg cells”. This sentence is incorrect and should be deleted. See for example: PMID: 28373295. Also I would move up the last paragraph of the discussion, it feels out of place.

Agreed. Per reviewer’s suggestion, we have reduced the length of the Discussion section by deleting the discussion on metabolic pathways as well as the last paragraph in the Discussion (a shorter version was put in the Figure legend). A number of other places were shortened as well. Finally, we deleted a sentence on “the first proteomic analysis on mouse

Treg cells” per reviewer’s suggestion, given a previous publication that the reviewer referred to.

All other Minor Comments have been addressed.

Thank you.

Reviewer #3 (Remarks to the Author):

The authors added important experiments to address reviewers, particularly the controls needed to confirm that the phenotypes result from loss of Rbx1. The data, as well as the mouse models, will be of great utility to the field. As such, I recommend publication in Nature Communications with the condition that the data (not only RNA-seq and microarray but also proteomics) and mice be made available to others upon publication. The authors should consider the following minor point when preparing the final version of their manuscript:

Thank you, and we have deposited all the data, including proteomics, and will make mice available to others upon request (Page 34, highlighted in green).

The introduction could still provide a more balanced view of the biochemical mechanisms but as the study largely focuses on immune functions I leave this to the authors’ discretion.

Thank you.

REVIEWERS' COMMENTS

Reviewer #1 (Remarks to the Author):

The authors have done everything possible to satisfy my requests. I think this is a very solid manuscript and although there are still major gaps in understanding, I think it is not fair to hold up publication any longer.

Reviewer #3 (Remarks to the Author):

The authors have performed a number of additional experiments to ensure the integrity of their conclusions.

I only have one remaining minor quibble: Our recent study showed that RBX1 binds to CDC34/UBCH2C to promote ubiquitylation of large number of substrates via the K48 linkage, whereas RBX2 binds to UBE2C/UBE2S to promote ubiquitylation of different set of substrates via the K11 linkage for proteasome degradation^{9,10}, demonstrating a new layer of difference between two family members.

Many other groups have demonstrated RBX1 binding to UBCH5B/UBE2D2 or UBCH5C/UBE2D3 and this should be noted. Also, since the original submission, Schulman published how RBX1 binds ARIH1 and RBX2 binds ARIH2. These works should be cited as demonstrating a new layer of difference between two family members.

REVIEWERS' COMMENTS

Reviewer #1 (Remarks to the Author):

The authors have done everything possible to satisfy my requests. I think this is a very solid manuscript and although there are still major gaps in understanding, I think it is not fair to hold up publication any longer.

Thanks for your positive comments.

Reviewer #3 (Remarks to the Author):

The authors have performed a number of additional experiments to ensure the integrity of their conclusions.

I only have one remaining minor quibble: Our recent study showed that RBX1 binds to CDC34/UBCH2C to promote ubiquitylation of large number of substrates via the K48 linkage, whereas RBX2 binds to UBE2C/UBE2S to promote ubiquitylation of different set of substrates via the K11 linkage for proteasome degradation^{9,10}, demonstrating a new layer of difference between two family members.

Many other groups have demonstrated RBX1 binding to UBCH5B/UBE2D2 or UBCH5C/UBE2D3 and this should be noted. Also, since the original submission, Schulman published how RBX1 binds ARIH1 and RBX2 binds ARIH2. These works should be cited as demonstrating a new layer of difference between two family members.

Thanks for your insight comments. These studies have been cited to give a more comprehensive demonstration of a new layer of difference between RBX1 and RBX2 (Page 4 & Page 37, highlighted).